# EFO$_k$-CQA: Towards Knowledge Graph Complex Query Answering beyond Set Operation

## Abstract

To answer complex queries on knowledge graphs, logical reasoning over incomplete knowledge is required due to the open-world assumption. Learning-based methods are essential because they are capable of generalizing over unobserved knowledge. Therefore, an appropriate dataset is fundamental to both obtaining and evaluating such methods under this paradigm. In this paper, we propose a comprehensive framework for data generation, model training, and method evaluation that covers the combinatorial space of Existential First-order Queries with multiple variables (EFO$_k$). The combinatorial query space in our framework significantly extends those defined by set operations in the existing literature. Additionally, we construct a dataset, EFO$_k$-CQA, with 741 query types for empirical evaluation, and our benchmark results provide new insights into how query hardness affects the results. Furthermore, we demonstrate that the existing dataset construction process is systematically biased that hinders the appropriate development of query-answering methods, highlighting the importance of our work. Our code and data are provided in `https://anonymous.4open.science/r/EFOK-CQA/README.md`.

## 1 Introduction

The Knowledge Graph (KG) is a powerful database that encodes relational knowledge into a graph representation [34, 31], supporting downstream tasks [41, 8] with essential factual knowledge. However, KGs suffer from incompleteness during its construction [34, 7], which is formally acknowledged as Open World Assumption (OWA) [19]. The task of Complex Query Answering (CQA) proposed recently has attracted much research interest [13, 28]. This task ambitiously aims to answer database-level complex queries described by logical complex connectives (conjunction $\wedge$, disjunction $\vee$, and negation $\neg$) and quantifiers[1] (existential $\exists$) [37, 27, 18]. However, CQA on KGs differs from query answering on databases in two aspects: (1) traditional query answering algorithms obtain incomplete answers because of the incomplete KG [13]; (2) the huge size of the knowledge graph limits the scalability of traditional algorithms [26]. Therefore, learning-based methods dominate the CQA tasks because they can empirically generalize to unseen knowledge as well as prevent the resource-demanding symbolic search.

The thriving of learning-based methods also puts an urgent request on high-quality datasets and benchmarks. In the previous study, datasets are developed by progressively expanding the **syntactical**

---

[1]The universal quantifier is usually not considered in query answering tasks, as a common practice from both CQA on KG [37, 27] and database query answering [25]

Submitted to the 37th Conference on Neural Information Processing Systems (NeurIPS 2023) Track on Datasets and Benchmarks. Do not distribute.

**expressiveness**, where conjunction [13], union [26], negation [28], and other operators [20] are taken into account sequentially. In particular, the dataset proposed in [28] contains all logical connectives and becomes the standard training set for model development. [36] proposed a large evaluation benchmark EFO-1-QA that systematically evaluates the combinatorial generalizability of CQA models on such queries. More related works are included in Appendix A.

However, the queries in aforementioned datasets [28, 36] are recently justified as "Tree-Form" queries [39] as they rely on the tree combinations of set operations. Compared to the well-established TPC-H decision support benchmark [25] for database query processing, queries in existing CQA benchmarks [28, 36] have two common shortcomings: (1) lack of **combinatorial answers**: only one variable is queried, and (2) lack of **structural hardness**: all existing queries subject to the structure-based tractability [29, 39]. It is rather questionable whether existing CQA data under such limited scope can support the future development of methodologies for general decision support with open-world knowledge.

The goal of this paper is to establish a new framework that addresses the aforementioned shortcomings to support further research in complex query answering on knowledge graphs. Our framework is formally motivated by the well-established investigation of constraint satisfaction problems, which all queries can be formulated as. In general, the contribution of our work is four folds.

**Complete coverage** We capture the complete Existential First Order (EFO) queries from their rigorous definitions, underscoring both **combinatorial hardness** and **structural hardness** and extending the existing coverage [36] which covers only a subset of $EFO_1$ query. The captured query family is denoted as $EFO_k$ where $k$ stands for multiple variables.

**Curated datasets** We derive $EFO_k$-CQA dataset, a non-exclusive extension of the previous EFO-1-QA benchmark [36] and contains 741 types of query. We design several rules to guarantee that our dataset includes high-quality nontrivial queries, particularly those that contain multiple query variables and are not structure-based tractable.

**Convenient implementation** We implement the entire pipeline for query generation, answer sampling, model training and inference, and evaluation for the undiscussed scenarios of **combinatorial answers**. Our pipeline is backward compatible, which supports both set operation-based methods and more recent ones.

**Results and findings** We evaluate six representative CQA methods on our benchmark. Our results refresh the previous empirical findings and further reveal the structural bias of previous data.

## 2 Problem definition

### 2.1 Existential first order (EFO) queries on knowledge graphs

Given a set $\mathcal{E}$ of entities and a set $\mathcal{R}$ of relations , a knowledge graph $\mathcal{KG}$ encodes knowledge as set of factual triple $\mathcal{KG} = \{(h, r, t)\} \subset \mathcal{E} \times \mathcal{R} \times \mathcal{E}$. According to the OWA, the knowledge graph that we have observed $\mathcal{KG}_o$ is only part of the real knowledge graph, meaning that $\mathcal{KG}_o \subset \mathcal{KG}$.

The existing research only focuses on the logical formulas without universal quantifiers [27, 35]. We then offer the definition of it based on strict first order logic.

**Definition 1** (Term). *A term is either a variable $x$ or an entity $a \in \mathcal{E}$.*

**Definition 2** (Atomic formula). *$\phi$ is an atomic formula if $\phi = r(h, t)$, where $r \in \mathcal{R}$ is a relation, $h$ and $t$ are two terms.*

**Definition 3** (Existential first order formula). *The set of the existential formulas is the smallest set $\Phi$ that satisfies the following:*

*(i) For atomic formula $r(h, t)$, itself and its negation $r(h, t), \neg r(h, t) \in \Phi$*

*(ii) If $\phi, \psi \in \Phi$, then $(\phi \wedge \psi), (\phi \vee \psi) \in \Phi$*

*(iii) If $\phi \in \Phi$ and $x_i$ is any variable, then $\exists x_i \phi \in \Phi$.*

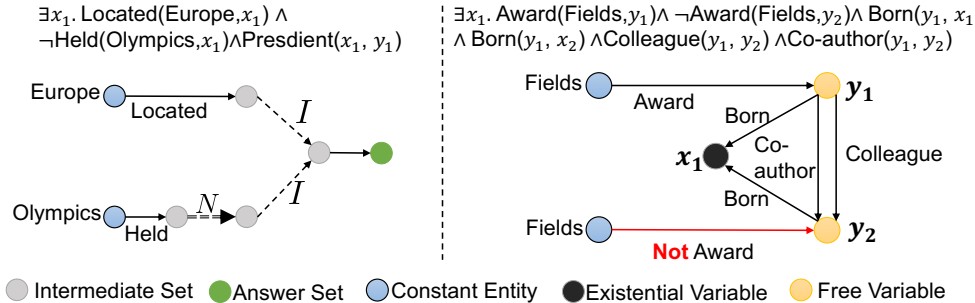

$\exists x_1.$ Located(Europe,$x_1$) $\wedge$
$\neg$Held(Olympics,$x_1$)$\wedge$Presdient($x_1$, $y_1$)

$\exists x_1.$ Award(Fields,$y_1$)$\wedge$ $\neg$Award(Fields,$y_2$)$\wedge$ Born($y_1$, $x_1$)
$\wedge$ Born($y_1$, $x_2$) $\wedge$Colleague($y_1$, $y_2$) $\wedge$Co-author($y_1$, $y_2$)

⬤ Intermediate Set 🟢 Answer Set 🔵 Constant Entity ⚫ Existential Variable 🟡 Free Variable

Figure 1: Operator Tree versus Query Graph. **Left**: An operator tree representing a given query "List the presidents of European countries that have never held the Olympics" [28]; **Right**: A query graph representing a given query "Find a pair of persons who are both colleagues and co-authors and were born in the same country, with one having awarded the fields medal while the another not", which is both a multigraph and a cyclic graph, containing two free variables.

**Definition 4** (Free variable). *If a variable $y$ is not associated with a quantifier, it is called a free variable, otherwise, it is called a bounded variable. We write $\phi(y_1, \cdots, y_k)$ to indicate $y_1, \cdots, y_k$ are the free variables of $\phi$.*

**Definition 5** (Sentence and query). *A formula $\phi$ is a sentence if it contains no free variable, otherwise, it is called a query. In this paper, we always consider formula with free variable, thus, we use formula and query interchangeably.*

**Definition 6** (Substitution). *For $a_1, \cdots, a_k$, where $a_i \in \mathcal{E}$, we write $\phi(a_1/y_1, \cdots, a_k/y_k)$ or simply $\phi(a_1, \cdots, a_k)$ for the result of simultaneously replacing all free occurrence of $y_i$ in $\phi$ by $a_i$, $i = 1, \cdots, k$.*

**Definition 7** (Answer of an EFO query). *For a given existential query $\phi(y_1, \cdots, y_k)$, its answer is a set that defined by*

$$\mathcal{A}[\phi(y_1, \cdots, y_k)] = \{(a_1, \cdots, a_k))|a_i \in \mathcal{E}, i = 1, \cdots, k, \phi(a_1, \cdots, a_k) \text{ is True}\}$$

**Definition 8** (Disjunctive Normal Form (DNF)). *For any existential formula $\phi(y_1, \cdots, y_k)$, it can be converted to the Disjunctive normal form as shown below:*

$$\phi(y_1, \cdots, y_k) = \gamma_1(y_1, \cdots, y_k) \vee \cdots \vee \gamma_m(y_1, \cdots, y_k) \tag{1}$$
$$\gamma_i(y_1, \cdots, y_k) = \exists x_1, \cdots, x_n.\rho_{i1} \wedge \cdots \wedge \rho_{it} \tag{2}$$

*where $\rho_{ij}$ is either an atomic formula or the negation of an atomic formula, $x_i$ is called an existential variable.*

DNF form has a strong property that $\mathcal{A}[\phi(y_1, \cdots, y_k)] = \cup_{i=1}^{m} \mathcal{A}[\gamma_i(y_1, \cdots, y_k)]$, which allows us to only consider conjunctive formulas $\gamma_i$ and then aggregate those answers to retrieve the final answers. This practical technique has been used in many previous research [22, 27]. Therefore, we only discuss conjunctive formulas in the rest of this paper.

## 2.2 Constraint satisfaction problem for EFO queries

Formally, a constraint satisfaction problem (CSP) $\mathcal{P}$ can be represented by a triple $\mathcal{P} = (X, D, C)$ where $X = (x_1, \cdots, x_n)$ is an $n$-tuple of variables, $D = (D_1, \cdots, D_n)$ is the corresponding $n$-tuple of domains, $C = (C_1, \cdots, C_t$ is $t$-tuple constraint, each constraint $C_i$ is a pair of $(S_i, R_{S_i})$ where $S_i$ is a set of variables $S_i = \{x_{i_j}\}$ and $R_{S_i}$ is the constraint over those variables [29].

Historically, there are strong parallels between CSP and conjunctive queries in knowledge bases [10, 17]. The terms correspond to the variable set $X$. The domain $D_i$ of a constant entity contains only itself, while it is the whole entity set $\mathcal{E}$ for other variables. Each constraint $C_i$ is binary that is induced by an atomic formula or its negation, for example, for an atomic formula $r(h, t)$, we have $S_i = \{h, t\}$, $R_{S_i} = \{(h, t)|h, t \in \mathcal{E}, (h, r, t) \in \mathcal{KG}\}$. Finally, by the definition of existential quantifier, we only consider the answer of free variable, rather than tracking all terms within the existential formulas.

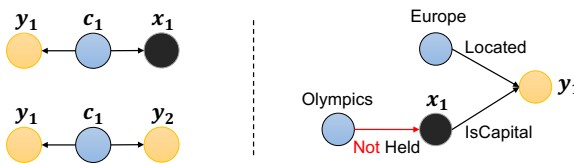

Figure 2: Left: Example of trivial abstract query graph, in the upper left graph, the $x_1$ is redundant violating Assumption 13, in the bottom left graph, answers for the whole query can be decomposed to answer two free variables $y_1$ and $y_2$ alone, violating Assumption 14. Right: Example of new query graph that is not included in previous benchmark [36] even though it can be represented by operator-tree. The representation of query graph follows Figure 1.

**Definition 9** (CSP answer of conjunctive formula). *For a conjunctive formula $\gamma$ in Equation 2 with $k$ free variables and $n$ existential variables, the answer set of it formulated as CSP instance is:*

$$\overline{\mathcal{A}}[\gamma(y_1, \cdots, y_k)] = \mathcal{A}[\gamma^{\star}(y_1, \cdots, y_{n+k})], \text{ where } \gamma^{\star} = \rho_{i1} \wedge \cdots \wedge \rho_{it}$$

This shows that the inference of existential formulas is easier than solving CSP instances since the existential variables do not need to be kept track of.

### 2.3 The representation of query

To give an explicit representation of existential formula, [13] firstly proposes to represent a formula by operator tree, where each node represents the answer set for a sub-query, and the logic operators in it naturally represent set operations. This method allows for the recursive computation from constant entity to the final answer set in a bottom-up manner [28]. However, this representation method is inherently directed, acyclic, and simple, therefore more recent research breaks these constraints by being bidirectional [21, 37] or being cyclic or multi [39]. To meet these new requirements, they propose to represent the formula by the query graph [39], which inherits the convention of constraint network in representing CSP instance. We utilize this design and further extend it to represent $\text{EFO}_k$ formula that contains multiple free variables. We provide the illustration and comparison of the operator tree and the query graph in Figure 1, where we show the strong expressiveness of the query graph. We also provide the formal definition of query graph as follows:

**Definition 10** (Query graph). *Let $\gamma$ be a conjunctive formula in equation 2, its query graph is defined by $G(\gamma) = \{(h, r, t, \{T, F\})\}$, where an atomic formula $\rho = r(h, t)$ in $\gamma$ corresponds to $(h, r, t, T)$ and $\rho = \neg r(h, t)$ corresponds to $(h, r, t, F)$.*

Therefore, any conjunctive formulas can be represented by a query graph, in the rest of the paper, we use query graphs and conjunctive formulas interchangeably.

## 3 The combinatorial space of $\text{EFO}_k$ queries

Although previous research has given a systematic investigation in the combinatorial space of operator trees [36], the combinatorial space of the query graph is much more challenging due to the extremely large search space and the lack of explicit recursive formulation. To tackle this issue on a strong theoretical background, we put forward additional assumptions to exclude trivial query graphs. Such assumptions or restrictions also exist in the previous dataset and benchmark [28, 36]. Specifically, we propose to split the task of generating data into two levels, the abstract level, and the grounded level. At the abstract level, we create *abstract query graph*, at the grounded level, we provide the abstract query graph with the relation and constant and instantiate it as a query graph. In this section, we elaborate on how we investigate the scope of the nontrivial $\text{EFO}_k$ query of interest step by step.

### 3.1 Nontrivial abstract query graph of $\text{EFO}_k$

The abstract query graph is the ungrounded query graph without information of certain knowledge graphs, and we give an example in Figure 3.

**Definition 11** (Abstract query graph). *The abstract query graph $\mathcal{G} = (V, E, f, g)$ is a directed graph with three node types,{**Constant Entity, Existential Variable, Free variable**}, and two edge types,{**positive, negative**}. The $V$ is the set of nodes, $E$ is the set of directed edges, $f$ is the function maps node to node type, $g$ is the function maps edge to edge type.*

**Definition 12** (Grounding). *For an abstract query graph $\mathcal{G}$, a grounding is a function $I$ that maps it into a query graph $I(\mathcal{G})$.*

We propose two assumptions of the abstract query graph as follows:

**Assumption 13** (No redundancy). *For a abstract query graph $\mathcal{G}$, there is not a subgraph $\mathcal{G}_s \subsetneqq \mathcal{G}$ such that for every grounding $I$, $\mathcal{A}[I(\mathcal{G})] = \mathcal{A}[I(\mathcal{G}_s)]$.*

**Assumption 14** (No decomposition). *For an abstract query graph $\mathcal{G}$, there are no such two subgraphs $\mathcal{G}_1$, $\mathcal{G}_2$, satisfying that $\mathcal{G}_1, \mathcal{G}_2 \subsetneqq \mathcal{G}$, such that for every instantiation $I$, $\mathcal{A}[I(\mathcal{G})] = \mathcal{A}[I(\mathcal{G}_1)] \times \mathcal{A}[I(\mathcal{G}_2)]$, where the $\times$ represents the Cartesian product.*

We note that the assumption 14 inherits the idea of the **structural** decomposition technique in CSP [11], which allows for solving a CSP instance by solving several sub-problems and combining the answer together based on topology property. Additionally, meeting these two assumptions in the grounded query graph is extremely computationally costly which we aim to avoid in practice.

We provide some easy examples to be excluded for violating the assumptions above in Figure 2.

### 3.2 Nontrivial query graph of EFO$_k$

Similarly, we propose two assumptions on the query graph.

**Assumption 15** (Meaningful negation). *For any negative edge $e$ in query graph $G$, we require removing it results in different CSP answers: $\overline{\mathcal{A}}[G - e] \neq \overline{\mathcal{A}}[G]$.*[2]

Assumption 15 treats negation separately because of the fact that for any $\mathcal{KG}$, any relation $r \in \mathcal{R}$, there is $|\{(h,t)|h,t \in \mathcal{E}, (h,r,t) \in \mathcal{KG}\}| \ll \mathcal{E}^2$, which means that the constraint induced by the negation of an atomic formula is much less "strict" than the one induced by a positive atomic formula.

**Assumption 16** (Appropriate answer size). *There is a constant $M \ll \mathcal{E}$ to bound the candidate set for each free variable $f_i$ in $G$, such that for any $i$, $|\{(a_i \in \mathcal{E}|(a_1, \cdots, a_i, \cdots, a_k) \in \mathcal{A}[G]\}| \leqslant M$.*

We note the Assumption 16 **extends** the "bounded negation" assumption in the previous dataset [28, 36]. We give an example "Find a city that is located in Europe and is the capital of a country that has not held the Olympics" in Figure 2, where the candidate set of $x_1$ is in fact bounded by its relation with the $y_1$ variable but not from the bottom "Olympics" constant, hence, this query is excluded in their dataset due to the directionality of operator tree.

Overall, the scope of the formula investigated in this paper surpasses the previous EFO-1-QA benchmark because of: (1). We include the EFO$_k$ formula with multiple free variables for the first time; (2). We include the whole family of EFO$_1$ query, many of them can not be represented by operator tree; (3) Our assumption is more systematic than previous ones as shown by the example in Figure 2. More details are offered in Appendix D.3.

## 4 Framework

We develop a versatile framework that supports five key functionalities fundamental to the whole CQA task: (1) Enumeration of nontrivial abstract query graphs as discussed in Section 3; (2) Sample grounding for the abstract query graph; (3) Compute answer for any query graph efficiently; (4) Support implementation of existing CQA models; (5) Conduct evaluation including newly introduced EFO$_k$ queries with multiple free variables. We explain each functionality in the following. An illustration of the first three functionalities is given in Figure 3.

---

[2]Ideally, we should expect them to have different answers as the existential formulas, however, this is computation costly and difficult to sample in practice, which is further discussed in Appendix D.

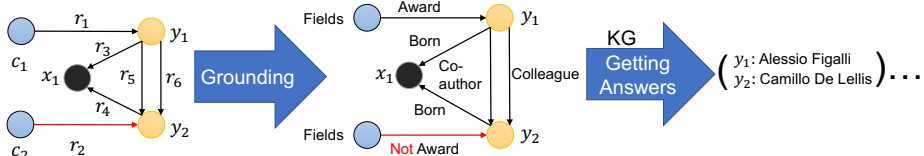

Figure 3: Illustration of the functionality of our framework. Left: abstract query graph, Middle: query graph, Right: answer of query.

### 4.1 Enumerate abstract query graph

As discussed in Section 3, we are able to abide by those assumptions as well as **enumerate** all possible query graphs within a given search space where certain parameters, including the number of constants, free variables, existential variables, and the number of edges are all given. Additionally, we apply the graph isomorphism algorithm to avoid duplicated query graphs being generated. More details for our generation method are provided in Appendix D.1.

### 4.2 Ground abstract query graph

To ground an abstract query graph $\mathcal{G}$ and comply with the assumption 15, we split the abstract query graph into two parts, the positive part and the negative part, $\mathcal{G} = \mathcal{G}_p \cup \mathcal{G}_n$. Then the grounding process is also split into two steps: 1. Sample grounding for the positive subgraph $\mathcal{G}_p$ and compute its answer 2. Ground the $\mathcal{G}_n$ to decrease the answer got in the first step. Details in Appendix D.2.

Finally, to fulfill the assumption 16, we follow the previous practice of manually filtering out queries that have more than 100 answers [28, 36], as we have introduced the $\text{EFO}_k$ queries, we slightly soften this constraint to be no more than $100 \times k$ answers.

### 4.3 Answer for existential formula

As illustrated in Section 2.2, the answer to an existential formula can be solved by a CSP solver, however, we also show in Definition 9 that CSP requires keeping track of the existential variables and it leads to huge computation costs. Thus, we develop our own algorithm following the standard solving technique of CSP, which ensures consistency conditions in the first step, and do the backtracking to get the final answers in the second step. Finally, we select part of our sampled queries and double-check it with the CSP solver https://github.com/python-constraint/python-constraint.

### 4.4 Learning-based methods

As the query graph is an extension to the operator tree regarding the express ability to existential formulas, we are able to reproduce CQA models that are initially implemented by the operator tree in our new framework. Specifically, since the operator tree is directed and acyclic, we compute its topology ordering that allows for step-by-step computation in the query graph. This algorithm is illustrated in detail in the Appendix F. We note our implementation coincides with the original one.

Conversely, for the newly proposed models that are based on query graphs, the original operator tree framework is not able to implement them, while our framework is powerful enough. We have therefore clearly shown that the query graph representation is more powerful than the previous operator tree and is able to support arbitrary existential formulas as explained in Section 2.3.

### 4.5 Evaluation protocol

As we have mentioned in Section 2.1, there is an observed knowledge graph $\mathcal{KG}_o$ and a full knowledge graph $\mathcal{KG}$. Thus, there is a set of observed answers $\mathcal{A}_o$ and a set of full answers $\mathcal{A}$ correspondingly. Since the goal of CQA is to tackle the challenge of OWA, it has been a common practice to evaluate CQA models by the "hard" answers $\mathcal{A}_h = \mathcal{A} - \mathcal{A}_o$ [26, 27]. However, to the best of our knowledge,

there has not been a systematic evaluation protocol for $EFO_k$ queries, thus we leverage this idea and propose three types of different metrics to fill the research gap in the area of evaluation of queries with multiple free variables, and thus have combinatorial answers.

**Marginal.** For any free variable $f_i$, its full answer is $\mathcal{A}^{f_i} = \{a_i \in \mathcal{E} | (a_1, \cdots, a_i, \cdots, a_k) \in \mathcal{A}\}$, the observed answer of it $\mathcal{A}_o^{f_i}$ is defined similarly. This is termed "solution projection" in CSP theory [12] to evaluate whether the locally retrieved answer can be extended to an answer for the whole problem. Then, we rank the hard answer $\mathcal{A}_h^{f_i} = \mathcal{A}^{f_i} - \mathcal{A}_o^{f_i}$ [3], against those non-answers $\mathcal{E} - \mathcal{A}^{f_i} - \mathcal{A}_o^{f_i}$ and use the ranking to compute standard metrics like MRR, HIT@K for every free variable. Finally, the metric on the whole query graph is taken as the average of the metric on all free variables. We note that this metric is an extension of the previous design proposed by [20]. However, this metric has the inherent drawback that it fails to evaluate the combinatorial answer by the $k$-length tuple and thus fails to find the correspondence among free variables.

**Multiply.** Because of the limitation of the marginal metric discussed above, we propose to evaluate the combinatorial answer by each $k$-length tuple $(a_1, \cdots, a_k)$ in the hard answer set $\mathcal{A}_h$. Specifically, we rank each $a_i$ in the corresponding node $f_i$ the same as the marginal metric. Then, we propose the HIT@$n^k$ metric, it is 1 if all $a_i$ is ranked in the top $n$ in the corresponding node $f_i$, and 0 otherwise.

**Joint.** Finally, we note these metrics above are not the standard way of evaluation, which is based on a joint ranking for all the $\mathcal{E}^k$ combinations of the entire search space. We propose to estimate the joint ranking in a closed form given certain assumptions, see Appendix E for the proof and details.

# 5  The $EFO_k$-CQA dataset and benchmark results

## 5.1  The $EFO_k$-CQA dataset

With the help of our framework developed in Section 4, we are able to develop a new dataset called $EFO_k$-CQA, whose combinatorial space is parameterized by the number of constants, existential and free variables, and the number of edges. $EFO_k$-CQA dataset includes 741 different abstract query graphs in total. The parameters and the generation process, as well as its statistics, are detailed in Appendix D.4.

Then, we conduct experiments on our new $EFO_k$-CQA dataset with six representative CQA models including BetaE [28], LogicE [24], and ConE [40], which are built on the operator tree, CQD [2], LMPNN [35], and FIT [39] which are built on query graph. The experiments are conducted in two parts, (1). the queries with one free variable, specifically, including those that can not be represented by operator tree; (2). the queries that contain multiple free variables.

We have made some adaptations to the implementation of CQA models, allowing them to infer $EFO_k$ queries, full detail is offered in Appendix F. The experiment is conducted on a standard knowledge graph FB15k-237 [32] and additional experiments on other standard knowledge graphs FB15k and NELL are presented in Appendix H.

## 5.2  Benchmark results for $k = 1$

Because of the great number of abstract query graphs, we follow [36] to group query graphs by three factors: (1). the number of constant entities; (2). the number of existential variables, and (3). the topology of the query graph[4]. The result is shown in Table 1.

**Structure analysis.** Firstly, we find a clear monotonic trend that adding constant entities makes a query easier while adding existing variables makes a query harder, which the previous research [36] fails to uncover. Besides, we are the first to consider the topology of query graphs: when the number

---

[3]We note $\mathcal{A}_h^{f_i}$ can be empty for some free variable or even for all free variables, making these marginal metrics not reliable, details in Appendix E.

[4]We make a further constraint in our $EFO_k$-CQA dataset that the total edge is at most as many as the number of nodes, thus, a graph can not be both a multigraph and a cyclic graph.

Table 1: HIT@10 scores(%) for inferring queries with one free variable on FB15k-237. We denote $e$ as the number of existential variables and $c$ as the number of constant entities. SDAG represents the Simple Directed Acyclic Graph, Multi for multigraph, and Cyclic for the cyclic graph. AVG.($c$) and AVG.($e$) is the average score of queries with the number of constant entities / existential variables fixed.

| Model | $c$ \ $e$ | 0 | 1 | | 2 | | | AVG.($c$) | AVG. |
|---|---|---|---|---|---|---|---|---|---|
| | | SDAG | SDAG | Multi | SDAG | Multi | Cyclic | | |
| BetaE | 1 | 31.4 | 33.0 | 22.3 | 21.1 | 17.7 | 30.7 | 22.1 | |
| | 2 | 57.2 | 36.2 | 35.5 | 29.3 | 29.4 | 45.3 | 32.5 | 36.4 |
| | 3 | 80.0 | 53.1 | 53.6 | 38.2 | 37.8 | 58.2 | 42.1 | |
| | AVG.($e$) | 59.3 | 43.8 | 40.6 | 33.8 | 32.7 | 49.3 | | |
| LogicE | 1 | 34.4 | 34.9 | 23.0 | 21.4 | 17.4 | 30.3 | 22.4 | |
| | 2 | 60.0 | 38.4 | 36.8 | 29.8 | 29.3 | 45.3 | 33.0 | 36.7 |
| | 3 | 83.0 | 55.5 | 55.5 | 38.5 | 37.8 | 57.8 | 42.4 | |
| | AVG.($e$) | 62.2 | 46.0 | 42.0 | 34.2 | 32.6 | 49.1 | | |
| ConE | 1 | 34.9 | 35.4 | 23.6 | 21.8 | 18.4 | 34.2 | 23.5 | |
| | 2 | 61.0 | 39.1 | 38.4 | 32.0 | 31.5 | 50.2 | 35.2 | 39.0 |
| | 3 | 84.8 | 56.7 | 57.1 | 41.1 | 40.0 | 63.4 | 44.9 | |
| | AVG.($e$) | 63.4 | 47.0 | 43.5 | 36.5 | 34.7 | 54.1 | | |
| CQD | 1 | **39.0** | 34.2 | 17.6 | 17.4 | 12.7 | 28.7 | 18.7 | |
| | 2 | 50.7 | 33.8 | 33.6 | 28.4 | 28.4 | 45.7 | 31.4 | 35.9 |
| | 3 | 58.4 | 49.6 | 52.4 | 39.3 | 39.1 | 60.4 | 42.6 | |
| | AVG.($e$) | 50.7 | 41.4 | 38.4 | 33.8 | 32.4 | 50.2 | | |
| LMPNN | 1 | 38.6 | 37.8 | 21.8 | 22.9 | 17.8 | 31.7 | 23.2 | |
| | 2 | 62.2 | 40.2 | 35.0 | 30.8 | 28.1 | 44.4 | 32.5 | 35.8 |
| | 3 | 86.6 | 56.9 | 51.9 | 38.3 | 35.3 | 55.8 | 40.8 | |
| | AVG.($e$) | 65.4 | 47.8 | 39.6 | 34.5 | 30.8 | 48.0 | | |
| FIT | 1 | 38.7 | **42.7** | **32.5** | **26.1** | **22.5** | **41.5** | **28.8** | |
| | 2 | **65.5** | **47.7** | **48.2** | **39.7** | **40.1** | **56.5** | **43.4** | **47.0** |
| | 3 | **84.2** | **63.9** | **63.5** | **50.5** | **50.4** | **63.5** | **53.6** | |
| | AVG.($e$) | **65.8** | **54.7** | **51.5** | **44.9** | **43.7** | **57.5** | | |

of constants and existential variables is fixed, we have found the originally investigated queries that correspond to Simple Directed Acyclic Graphs (SDAG) are generally easier than the multigraphs ones but harder than the cyclic graph ones. This is an intriguing result that greatly deviates from traditional CSP theory in close world which finds that the cyclic graph is NP-complete, while the acyclic graph is tractable [6]. Our conjecture for this intriguing result in the open world is that the cyclic graph contains one more constraint than SDAG that serves as a source of information for CQA models, while the multigraph tightens an existing constraint and thus makes the query harder.

**Model analysis.** For models that are built on operator tree, including BetaE, LogicE, and ConE, their relative performance is steady among all breakdowns and is consistent with their reported score in the original dataset [28], showing similar generalizability. However, for models that are built on query graphs, including CQD, LMPNN, and FIT, we have found that LMPNN performs generally better than CQD in SDAG, but falls behind CQD in multigraphs and cyclic graphs. We assume the reason behind this is that LMPNN requires training while CQD does not, however, the original dataset are **biased** which only considers SDAG, leading to the result that LMPNN doesn't generalize well to the unseen tasks with different topology property. We expect future CQA models may use our framework to address this issue of biased data and generalize better to more complex queries.

We note FIT is designed to infer all $EFO_1$ queries and is indeed able to outperform other models in almost all breakdowns, however, its performance comes with the price of computational cost, and

Table 2: HIT@10 scores(%) of three different types for answering queries with two free variables on FB15k-237. The constant number is fixed to be two. $e$ is the number of existential variables. The SDAG, Multi, and Cyclic are the same as Table 1.

| Model | HIT@10 Type | $e = 0$ | | $e = 1$ | | | $e = 2$ | | | AVG. |
|---|---|---|---|---|---|---|---|---|---|---|
| | | SDAG | Multi | SDAG | Multi | Cyclic | SDAG | Multi | Cyclic | |
| BetaE | Marginal | 54.5 | 50.2 | 49.5 | 46.0 | 58.8 | 37.2 | 35.5 | 58.3 | 43.8 |
| | Multiply | 27.3 | 22.4 | 22.3 | 16.9 | 26.2 | 16.9 | 13.9 | 25.7 | 18.3 |
| | Joint | 6.3 | 5.4 | 5.2 | 4.2 | 10.8 | 2.2 | 2.3 | 9.5 | 4.5 |
| LogicE | Marginal | 58.2 | 50.9 | 52.2 | 47.4 | 60.4 | 37.7 | 35.8 | 59.2 | 44.6 |
| | Multiply | 32.1 | 23.1 | 24.9 | 18.1 | 28.3 | 18.1 | 14.8 | 26.6 | 19.5 |
| | Joint | 6.8 | 6.0 | 6.1 | 4.5 | 12.3 | 2.5 | 2.7 | 10.3 | 5.1 |
| ConE | Marginal | 60.3 | 53.8 | 54.2 | 50.3 | **66.2** | 40.1 | 38.5 | **63.7** | 47.7 |
| | Multiply | 33.7 | 25.2 | 26.1 | 19.8 | 32.1 | 19.5 | 16.3 | 30.3 | 21.5 |
| | Joint | 6.7 | 6.4 | 6.2 | 4.8 | 12.6 | 2.6 | 2.7 | 10.9 | 5.3 |
| CQD | Marginal | 50.4 | 46.5 | 49.1 | 45.6 | 59.7 | 33.5 | 33.1 | 61.5 | 42.8 |
| | Multiply | 28.9 | 23.4 | 25.4 | 19.5 | 31.3 | 17.8 | 16.0 | 30.5 | 21.0 |
| | Joint | **8.0** | 8.0 | 7.4 | 6.0 | **13.9** | 3.6 | 3.9 | **12.0** | **6.4** |
| LMPNN | Marginal | 58.4 | 51.1 | 54.9 | 49.2 | 64.7 | 39.6 | 36.1 | 58.7 | 45.4 |
| | Multiply | 35.0 | 26.7 | 29.2 | 21.7 | **33.4** | 21.4 | 17.0 | 28.4 | 22.2 |
| | Joint | 7.6 | 7.5 | 7.1 | 5.3 | 12.9 | 2.8 | 2.9 | 9.5 | 5.2 |
| FIT | Marginal | **64.3** | **61.0** | **63.1** | **60.7** | 58.5 | **49.0** | **49.1** | 60.2 | **54.3** |
| | Multiply | **39.7** | **32.2** | **35.9** | **27.8** | 27.4 | **29.5** | **26.8** | **32.4** | **29.2** |
| | Joint | 7.4 | **9.0** | **7.8** | **6.5** | 10.1 | **3.7** | **4.6** | 10.6 | **6.4** |

face challenges in cyclic graph where it degenerates to enumeration: which we further explain in Appendix F.

## 5.3 Benchmark results for $k = 2$

As we have explained in Section 4.5, we propose three kinds of metrics, marginal ones, multiply ones, and joint ones, from easy to hard, to evaluate the performance of a model in the scenario of multiple variables. The evaluation result is shown in Table 2. As the effect of the number of constant variables is quite clear, we remove it and add the metrics based on HIT@10 as the new factor.

For the impact regarding the number of existential variables and the topology property of the query graph, we find the result is similar to Table 1, which may be explained by the fact that those models are all initially designed to infer queries with one free variable. For the three metrics we have proposed, we have identified a clear difficulty difference among them though they generally show similar trends. The scores of joint HIT@10 are pretty low, indicating the great hardness of answering queries with multiple variables. Moreover, we have found that FIT falls behind other models in some breakdowns which are mostly cyclic graphs, corroborating our discussion in Section 5.2.

## 6 Conclusion

In this paper, we make a thorough investigation of the family of $\text{EFO}_k$ formulas based on strong theoretical background. We then present a new powerful framework that supports several functionalities essential to CQA task, with this help, we build the $\text{EFO}_k$-CQA dataset that greatly extends the previous dataset and benchmark. Our evaluation result brings new empirical findings and reflects the biased selection in the previous dataset impairs the performance of CQA models, emphasizing the contribution of our work.

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
