## A  Related works

Answering complex queries on knowledge graphs differs from database query answering by being a data-driven task [37], where the open-world assumption is addressed by methods that learn from data. Meanwhile, learning-based methods enable faster neural approximate solutions of symbolic query answering problems [27].

The prevailing way is query embedding, where the computational results are embedded and computed in the low-dimensional embedding space. Specifically, the query embedding over the set operator trees is the earliest proposed [13]. The supported set operators include projection[13], intersection [26], union and negation [28], and later on be improved by various designs [40, 3]. Such methods assume queries can be converted into the recursive execution of set operations, which imposes additional assumptions on the solvable class of queries [36]. These assumptions introduce additional limitations of such query embeddings

Recent advancements in query embedding methods adapt query graph representation and graph neural networks, supporting atomics [21] and negated atomics [35]. Query embedding on graphs bypasses the assumptions for queries [36]. Meanwhile, other search-based inference methods [2, 39] are rooted in fuzzy calculus and not subject to the query assumptions [36].

Though many efforts have been made, the datasets of complex query answering are usually subject to the assumptions by set operator query embeddings [36]. Many other datasets are proposed to enable queries with additional features, see [27] for a comprehensive survey of datasets. However, only one small dataset proposed by [39] introduced queries and answers beyond such assumptions [36]. It is questionable that this small dataset is fair enough to justify the advantages claimed in advancement methods [35, 39] that aim at complex query answering. The dataset [39] is still far away from the systematical evaluation as [36] and $\text{EFO}_k$-CQA proposed in this paper fills this gap.

## B  Details of constraint satisfaction problem

In this section, we introduce the constraint satisfaction problem (CSP) again. One instance of CSP $\mathcal{P}$ can be represented by a triple $\mathcal{P} = (X, D, C)$ where $X = (x_1, \cdots, x_n)$ is an $n$-tuple of variables, $D = (D_1, \cdots, D_n)$ is the corresponding $n$-tuple of domains, meaning for each $i$, $x_i \in D_i$. Then, $C = (C_1, \cdots, C_t)$ is $t$-tuple constraint, each constraint $C_i$ is a pair of $(S_i, R_{S_i})$ where $S_i$ is called the scope of the constraint, meaning it is a set of variables $S_i = \{x_{i_j}\}$ and $R_{S_i}$ is the constraint over those variables [29], meaning that $R_{S_i}$ a subset of the cartesian product of variables in $S_i$.

Then the formulation of existential conjunctive formulas as CSP has already been discussed in Section 2.2. Additionally, for the negation of atomic formula $\neg r(h, t)$, we note the constraint $C$ is also binary with $S_i = \{h, t\}$, $R_{S_i} = \{(h, t) | h, t \in \mathcal{E}, (h, r, t) \notin \mathcal{KG}\}$, this means that $R_{S_i}$ is a very large set, thus the constraint is less "strict" than the positive ones.

## C  Preliminary of tree form query

We explain the operator tree method, as well as the tree-form queries in this section, which is firstly introduced in [39]. The tree-form queries are defined to be the syntax closure of the operator tree method and are the prevailing query types in the existing datasets [28, 36], see the definition below:

**Definition 17** (Tree-Form Query)**.** *The set of the Tree-Form queries is the smallest set $\Phi$ such that:*

    *(i) If $\phi(y) = r(a, y)$, where $a \in \mathcal{E}$, then $\phi(y) \in \Phi$;*
    *(ii) If $\phi(y) \in \Phi$, $\neg\phi(y) \in \Phi$;*
    *(iii) If $\phi(y), \psi(y) \in \Phi$, then $(\phi \wedge \psi)(y) \in \Phi$ and $(\phi \vee \psi)(y) \in \Phi$;*
    *(iv) If $\phi(y) \in \Phi$ and $y'$ is any variable, then $\psi(y') = \exists y.r(y, y') \wedge \phi(y) \in \Phi$.*

We note that the family of tree-form queries deviates from the targeted $\text{EFO}_1$ query family [39]. The rationale of the definition is that the previous model relied on the representation of "**operator tree**"

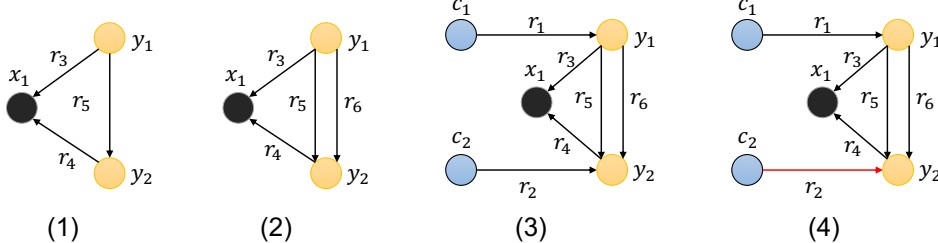

Figure 4: The four steps of enumerating the abstract query graphs. We note that the example and representation follow Figure 3.

which addresses logical queries to simulate logical reasoning as the execution of set operators [28, 40, 38], where each node represents a set of entities corresponding to the answer set of a sub-query [39]. Then, logical connectives are transformed into operator nodes for set projections(Definition 17 i,iv), complement(Definition 17 ii), intersection, and union(Definition 17 iii) [36]. Particularly, the set projections are derived from the Skolemization of predicates [24]. Therefore, the operator tree method that has been adopted in lines of research [28, 40, 38] is just a model that neuralizes these set operations: projection, complement, intersection, and union. These different models basically only differ from each other by their parameterization while having the same expressiveness as characterized by the tree form query.

Specifically, the left side of the Figure 1 shows an example of the operator tree, where "Held" and "Located" are treated as two projections, "N" represents set complement, and "I" represents set intersection. Therefore, the embedding of the root representing the answer set can be computed based on these set operations in a bottom-up manner [28].

Finally, it has been noticed that tree-form query is subject to structural traceability and only has polynomial time combined complexity for inference while the general $EFO_k$, or even $EFO_1$ queries, is NP-complete, with detailed proof in [39]. Therefore, this result highlights the importance of investigating the $EFO_k$ queries as it greatly extends the previous tree-form queries.

# D  Construction of the whole $EFO_k$-CQA datset

In this section, we provide details for the construction of the $EFO_k$-CQA dataset.

## D.1  Enumeration of the abstract query graphs

We first give a proposition of the property of abstract query graph:

**Proposition 18.** *For an abstract query graph $\mathcal{G}$, if it conforms Assumption 13 and Assumption 14, then removing all constant entities in $\mathcal{G}$ will lead to only one connected component and no edge is connected between two constant entities.*

*Proof.* We prove this by contradiction. If there is an edge (whether positive or negative) between constant entities, then this edge is redundant, violating Assumption 13. Then, if there is more than one connected component after removing all constant entities in $\mathcal{G}$. Suppose one connected component has no free variable, then this part is a sentence and thus has a certain truth value, whether 0 or 1, which is redundant, violating Assumption 13. Then, we assume every connected component has at least one free variable, we assume there is $m$ connected component and we have:

$$Node(\mathcal{G}) = (\cup_{i=1}^{m} Node(\mathcal{G}_i)) \cup Node(\mathcal{G}_c)$$

where $m > 1$, the $\mathcal{G}_c$ is the set of constant entities and each $\mathcal{G}_i$ is the connected component, we use $Node(\mathcal{G})$ to denote the node set for a graph $\mathcal{G}$. Then this equation describes the partition of the node set of the original $\mathcal{G}$.

Then, we construct $\mathcal{G}_a = G[Node(\mathcal{G}_1) \cup \mathcal{G}_c]$ and $\mathcal{G}_b = G[(\cup_{i=1}^m Node(\mathcal{G}_i)) \cup Node(\mathcal{G}_c)]$, where $G$ represents the induced graph. Then we naturally have that $\mathcal{A}[I(\mathcal{G})] = \mathcal{A}[I(\mathcal{G}_a)] \times \mathcal{A}[I(\mathcal{G}_b)]$, where the $\times$ represents the Cartesian product, violating Assumption 14.

$\square$

Additionally, as mentioned in Appendix B, the negative constraint is less "strict", we formally put an additional assumption of the real knowledge graph as the following:

**Assumption 19.** *For any knowledge graph $\mathcal{KG}$, with its entity set $\mathcal{E}$ and relations set $\mathcal{R}$, we assume it is somewhat sparse with regard to each relation, meaning: for any $r \in \mathcal{R}, |\{a \in \mathcal{E}|\exists b.(a, r, b) \in \mathcal{KG} \text{ or } (b, r, a) \in \mathcal{KG}\}| \ll \mathcal{E}$*

Then we develop another proposition for the abstract query graph:

**Proposition 20.** *With the knowledge graph conforming Assumption 19, for any node $u$ in the abstract query graph $\mathcal{G}$, if $u$ is an existential variable or free variable, then it can not only connect with negative edges.*

*Proof.* Suppose $u$ only connects to $m$ negative edge $e_1, \cdots, e_m$. For any grounding $I$, we assume $I(e_i) = r_i \in \mathcal{R}$. For each $r_i$, we construct its endpoint set

$$\text{Endpoint}(r_i) = \{a \in \mathcal{E}|\exists b.(a, r, b) \in \mathcal{KG} \text{ or } (b, r, a) \in \mathcal{KG}\}$$

by the assumption 19, we have $|Endpoint(r_i)| \ll \mathcal{E}$, then we have:

$$|\cup_{i=1}^m \text{Endpoint}(r_i)| \leqslant \Sigma_{i=1}^m |\text{Endpoint}(r_i)| \ll \mathcal{E}$$

since $m$ is small due to the size of the abstract query graph. Then we have two situations about the type of node $u$:

**1.If node $u$ is an existential variable.**

Then we construct a subgraph $\mathcal{G}_s$ be the induced subgraph of $Node(\mathcal{G}) - u$, then for any possible grounding $I$, we prove that $\mathcal{A}[I(\mathcal{G}_s)] = \mathcal{A}[I(\mathcal{G})]$, the right is clearly a subset of the left due to it contains more constraints, then we show every answer of the left is also an answer on the right, we merely need to give an appropriate candidate in the entity set for node $v$, and in fact, we choose any entity in the set $\mathcal{E} - \cup_{i=1}^m \text{Endpoint}(r_i)$ since it suffices to satisfies all constraints of node $u$, and we have proved that $|\mathcal{E} - \cup_{i=1}^m \text{Endpoint}(r_i)| > 0$.

This violates the Assumption 13.

**2.If node $u$ is a free variable.**

Similarly, any entity in the set $\mathcal{E} - \cup_{i=1}^m \text{Endpoint}(r_i)$ will be an answer for the node $u$, thus violating the Assumption 16.

$\square$

We note the proposition 20 extends the previous requirement about negative queries, which is firstly proposed in [28] and inherited and named as "bounded negation" in [36], the "bounded negation" requires the negation operator should be followed by the intersection operator in the operator tree. Obviously, the abstract query graph that conforms to "bounded negation" will also conform to the requirement in Proposition 20. A vivid example is offered in Figure 2.

Finally, we make the assumption of the distance to the free variable of the query graph:

**Assumption 21.** *There is a constant $d$, such that for every node $u$ in the abstract query graph $\mathcal{G}$, it can find a free variable in its $d$-hop neighbor.*

We have this assumption to exclude the extremely long-path queries.

Equipped with the propositions and assumptions above, we explore the combinatorial space of the abstract query graph given certain hyperparameters, including: the max number of free variables,

max number of existential variables, max number of constant entities, max number of all nodes, max number of all edges, max number of edges surpassing the number of nodes, max number of negative edge, max distance to the free variable. In practice, these numbers are set to be: 2, 2, 3, 6, 6, 0, 1, 3. We note that the max number of edges surpassing the number of nodes is set to 0, which means that the query graph can at most have one more edge than a simple tree, thus, we exclude those query graphs that are both cyclic graphs and multigraphs, making our categorization and discussion in the experiments in Section 5.2 and Section 5.3 much more straightforward and clear.

Then, we create the abstract query graph by the following steps, which is a graph with three types of nodes and two kinds of edges:

1. First, create a simple connected graph $\mathcal{G}_1$ with two types of nodes, the existential variable and the free variable, and one type of edge, the positive edge.

2. We add additional edges to the simple graph $\mathcal{G}_1$ and make it a multigraph $\mathcal{G}_2$.

3. Then, the constant variable is added to the graph $\mathcal{G}_2$, In this step, we make sure not too long existential leaves. The result is graph $\mathcal{G}_3$.

4. Finally, random edges in $\mathcal{G}_3$ are replaced by the negation edge, and we get the final abstract query graph $\mathcal{G}_4$.

In this way, all possible query graphs within a certain combinatorial space are enumerated, and finally, we filter duplicated graphs with the help of the graph isomorphism algorithm. We give an example to illustrate the four-step construction of an abstract query graph in Figure 4.

### D.2 Ground abstract query graph with meaningful negation

To fulfill the Assumption 15 as discussed in Section 4.2, for an abstract query graph $\mathcal{G} = (V, E, f, g)$, we have two steps: (1). Sample grounding for the positive subgraph $\mathcal{G}_p$ and compute its answer (2). Ground the $\mathcal{G}_n$ to decrease the answer got in the first step. Then we define positive subgraph $\mathcal{G}_p$ to be defined as such, its edge set $E' = \{e \in E | g(e) = positive\}$, its node set $V' = \{u | u \in V, \exists e \in E'$ and $e$ connects to $u\}$. Then $\mathcal{G}_p = (V', E', f, g)$. We note that because of Proposition 20, if a node $u \in V - V'$, then we know node $u$ must be a constant entity.

Then we sample the grounding for the positive subgraph $\mathcal{G}_p$, we also compute the CSP answer $\overline{\mathcal{A}}_p$ for this subgraph.

Then we ground what is left in the positive subgraph, we split each negative edge in $E - E'$ into two categories:

**1. This edge $e$ connects two nodes $u, v$, and $u, v \in V'$.**

In this case, we sample the relation $r$ to be the grounding of $e$ such that it negates some of the answers in $\overline{\mathcal{A}}_p$.

**2. This edge $e$ connects two nodes $u, v$, where $u \in V'$, while $v \notin V'$.**

In this case, we sample the relation $r$ for $e$ and entity $a$ for $v$ such that they negate some answer in $\overline{\mathcal{A}}_p$, we note we only need to consider the possible candidates for node $u$ and it is quite efficient.

We note that there is no possibility that neither of the endpoints is in $V'$ because as we have discussed above, this means that both nodes are constant entities, but in Proposition 18 we have asserted that no edge is connected between two entities.

### D.3 The comparison to previous benchmark

To give an intuitive comparison of our EFO$_k$-CQA dataset against those previous datasets and benchmark, including the BetaE dataset in [28], the EFO-1-QA benchmark [36] that extends BetaE dataset, and the FIT dataset in [39] that explores 10 more new query types, we offer a new figure in Figure 5.

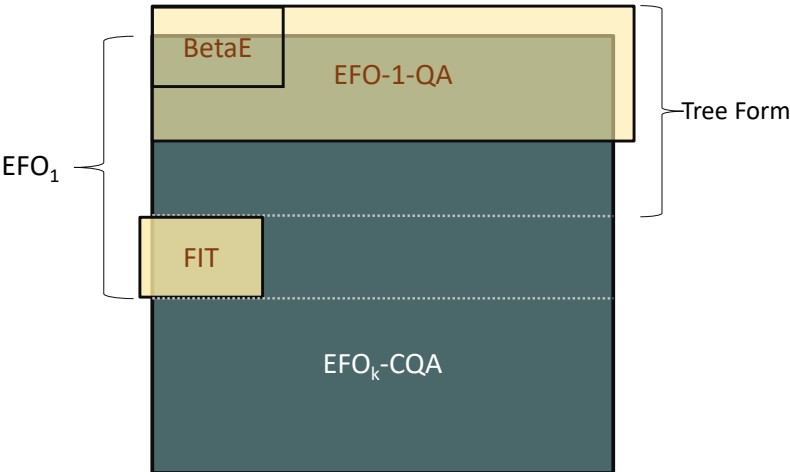

Figure 5: Illustration of the comparison between the $EFO_k$-CQA dataset (navy blue box) and the previous dataset (three yellow boxes), where the BetaE and EFO-1-QA aim to investigate the tree form query, explained in Appendix C, while the FIT dataset aims to investigate $EFO_1$ query that is not tree form. FIT is not a subset of $EFO_k$-CQA because its "3pm" query is not included in $EFO_k$-CQA.

It can be clearly observed that EFO-1-QA covers the BetaE dataset and has provided a quite systematic investigation in tree form query, while FIT deviates from them and studies ten new query types that are in $EFO_1$ but not tree form.

As discussed in Section 3, the scope of the formula investigated in our $EFO_k$-CQA dataset surpasses the previous EFO-1-QA benchmark and FIT dataset because of three reasons: (1). We include the $EFO_k$ formula with multiple free variables that has never been investigated(the bottom part of navy blue box in Figure 5); (2). We systematically investigate those $EFO_1$ queries that are not tree form while the previous FIT dataset only discusses ten hand-crafted query types (the navy blue part between two white lines in Figure 5); (3) Our assumption is more systematic than previous ones as shown by the example in Figure 2(the top navy blue part above two white lines in Figure 5). Though we only contain 741 query types while the EFO-1-QA benchmark contains 301 query types, we list reasons for the number of query types is not significantly larger than the previous benchmark: (1). EFO-1-QA benchmark relies on the operator tree that contains union, which represents the logic conjunction($\vee$), however, we only discuss the conjunctive queries because we always utilize the DNF of a query. We notice that there are only 129 query types in EFO-1-QA without the union, significantly smaller than the $EFO_k$-CQA dataset. (2). In the construction of $EFO_k$-CQA dataset, we restrict the query graph to have at most one negative edge to avoid the total number of query types growing quadratically, while in EFO-1-QA benchmark, their restrictions are different than ours and it contains queries that have two negative atomic formulas as indicated by the right part of yellow box is not contained in the navy blue box.

## D.4 $EFO_k$-CQA statistics

The statistics of our $EFO_k$-CQA dataset are shown in Table 3 and Table 4, they show the statistics of our abstract query graph by their topology property, the statistics are split into the situation that the number of free variable $k = 1$ and the number of free variable $k = 2$, correspondingly. We note abstract query graphs with seven nodes have been excluded as the setting of hyperparameters discussed in Appendix D.1, we make these restrictions to control the quadratic growth in the number of abstract query graphs.

Finally, in FB15k-237, we sample 1000 queries for an abstract query graph without negation, 500 queries for an abstract query graph with negation; in FB15k, we sample 800 queries for an abstract

Table 3: The number of abstract query graphs with one free variable. We denote $e$ as the number of existential variables and $c$ as the number of constant entities. SDAG represents the Simple Directed Acyclic Graph, Multi for multigraph, and Cyclic for the cyclic graph. Sum.($c$) and Sum.($e$) is the total number of queries with the number of constant entities / existential variables fixed.

| $c$ \ $e$ | 0 | 1 | | 2 | | | Sum.($c$) | Sum. |
|---|---|---|---|---|---|---|---|---|
| | SDAG | SDAG | Multi | SDAG | Multi | Cyclic | | |
| 1 | 1 | 2 | 4 | 4 | 16 | 4 | 31 | |
| 2 | 2 | 6 | 6 | 20 | 40 | 8 | 82 | 251 |
| 3 | 2 | 8 | 8 | 36 | 72 | 12 | 138 | |
| Sum.($e$) | 5 | 16 | 18 | 60 | 128 | 24 | | |

Table 4: The number of abstract query graphs with two free variables. The notation of $e$, $c$ SDAG, Multi, and Cyclic are the same as Table 3. And "-" means that this type of abstract query graph is not included.

| $c$ \ $e$ | $e = 0$ | | $e = 1$ | | | $e = 2$ | | | AVG. |
|---|---|---|---|---|---|---|---|---|---|
| | SDAG | Multi | SDAG | Multi | Cyclic | SDAG | Multi | Cyclic | |
| $c = 1$ | 1 | 2 | 7 | 18 | 4 | 6 | 32 | 26 | 96 |
| $c = 2$ | 4 | 4 | 20 | 36 | 8 | 38 | 108 | 64 | 282 |
| $c = 3$ | 4 | 4 | 32 | 60 | 12 | - | - | - | 112 |

640 query graph without negation, 400 queries for an abstract query graph with negation; in NELL,
641 we sample 400 queries for an abstract query graph without negation, 100 queries for an abstract
642 query graph with negation. As we have discussed in Appendix D.2, sample negative query is
643 computationally costly, thus we sample less of them.

# E  Evaluation details

645 We explain the evaluation protocol in detail for Section 4.5.

646 Firstly, we explain the computation of common metrics, including Mean Reciprocal Rank(MRR) and
647 HIT@K, given the full answer $\mathcal{A}$ in the whole knowledge graph and the observed answer $\mathcal{A}_o$ in the
648 observed knowledge graph, we focus on the hard answer $\mathcal{A}_h$ as it requires more than memorizing the
649 observed knowledge graph and serves as the indicator of the capability of reasoning.

650 Specifically, we rank each hard answer $a \in \mathcal{A}_h$ against all non-answers $\mathcal{E} - \mathcal{A} - \mathcal{A}_o$, the reason is
651 that we need to neglect other answers so that answers do not interfere with each other, finally, we get
652 the ranking for $a$ as $r$. Then its MRR is $1/r$, and its HIT@k is $\mathbf{1}_{r \leqslant k}$, thus, the score of a query is the
653 mean of the scores of every its hard answer. We usually compute the score for a query type (which
654 corresponds to an abstract query graph) as the mean score of every query within this type.

655 As the marginal score and the multiply score have already been explained in Section 4.5, we only
656 mention one point that it is possible that every free variable does not have marginal hard answer.
657 Assume that for a query with two free variables, its answer set $\mathcal{A} = \{(a_1, a_2), (a_1, a_3), (a_4, a_2)\}$ and
658 its observed answer set $\mathcal{A}_o = \{(a_1, a_3), (a_4, a_2)\}$. In this case, $a_1$ is not the marginal hard answer for
659 the first free variable and $a_2$ is not the marginal hard answer for the second free variable, in general,
660 no free variable has its own marginal hard answer.

661 Then we only discuss the joint metric, specifically, we only explain how to estimate the joint ranking
662 by the individual ranking of each free variable. For each possible $k$-tuple $(a_1, \cdots, a_k)$, if $a_i$ is ranked
663 as $r_i$ among the **whole** entity set $\mathcal{E}$, we compute the score of this tuple as $\Sigma_{i=1}^{k} r_i$, then we sort
664 the whole $\mathcal{E}^k$ $k$-tuple by their score, for the situation of a tie, we just use the lexicographical order.
665 After the whole joint ranking is got, we use the standard evaluation protocol that ranks each hard

---

**Algorithm 1** Embedding computation on the query graph.

---

**Require:** The query graph $G$.
  Compute the ordering of the nodes as explained in Algorithm 2.
  Create a dictionary $E$ to store the embedding for each node in the query graph
  **for** $i \leftarrow 1$ to $n$ **do**
    **if** node $u_i$ is a constant entity **then**
      The embedding of $u_i$, $E[i]$ is gotten from the entity embedding
    **else**
      Then we know node $u_i$ is either free variable or existential variable
      Compute the set of nodes $\{u_{i_j}\}_{j=1}^t$ that are previous to $i$ and adjacency to node $u_i$.
      Create a list to store projection embedding $L$.
      **for** $j \leftarrow 1$ to $t$ **do**
        Find the relation $r$ between node $u_i$ and $u_{i_j}$, get the embedding of node $u_{i_j}$ as $E[i_j]$.
        **if** $E[i_j]$ is not None **then**
          **if** The edge between $u_i$ and $u_{i_J}$ is positive **then**
            Compute the embedding of projection($E[i_j], r$), add it to the list $L$.
          **else**
            Compute the embedding of the negation of the projection($E[i_j], r$), add it to the list $L$.
          **end if**
        **end if**
      **end for**
      **if** The list $L$ has no element **then**
        $E[i]$ is set to none.
      **else if** The list $L$ has one element **then**
        $E[i] = L[0]$
      **else**
        Compute the embedding as the intersection of the embedding in the list $L$, and set $E[i]$ as the outcome.
      **end if**
    **end if**
  **end for**
  **return** The embedding dictionary $E$ for each node in the query graph.

---

answer against all non-answers. It can be confirmed that this estimation method admits a closed-form solution for the sorting in $\mathcal{E}^k$ space, thus the computation cost is affordable.

We just give the closed-form solution when there are two free variables:

for the tuple $(r_1, r_2)$, the possible combinations that sum less than $r_1 + r_2$ is $\binom{r_1+r_2-1}{2}$, then, there is $r_1 - 1$ tuple that ranks before $(r_1, r_2)$ because of lexicographical order, thus, the final ranking for the tuple $(r_1, r_2)$ is just $\binom{r_1+r_2-1}{2} + r_1$ that can be computed efficiently.

# F Implementation details of CQA models

In this section, we provide implementation details of CQA models that have been evaluated in our paper. For query embedding methods that rely on the operator tree, including BetaE [28], LogicE [24], and ConE [40], we compute the ordering of nodes in the query graph in Algorithm 2, then we compute the embedding for each node in the query graph Algorithm 1, the final embedding of every free node are gotten to be the predicted answer. Especially, the node ordering we got in Algorithm 2 coincides with the natural topology ordering induced by the directed acyclic operator tree, so we can compute the embedding in the same order as the original implementation. Then, in Algorithm 1, we implement each set operation in the operator tree, including intersection, negation, and set projection. By the merit of the Disjunctive Normal Form (DNF), the union is tackled in the final step. Thus, our implementation can coincide with the original implementation in the original dataset [28].

**Algorithm 2** Node ordering on the abstract query graph.
***
**Require:** The abstract query graph $\mathcal{G} = (V, E, f, g)$, $V$ consists $m$ nodes, $u_1, \cdots, u_m$.
    Creates an empty list $L$ to store the ordering of the node.
    Creates another two set $S_1$ and $S_2$ to store the nodes that are to be explored next.
    **for** $i \leftarrow 1$ to $m$ **do**
        **if** The type of node $f(u_i)$ is constant entity **then**
            list $L$ append the node $u_i$
            **for** Node $u_j$ that connects to $u_i$ **do**
                **if** $f(u_j)$ is existential variable **then**
                    $u_j$ is added to set $S_1$
                **else**
                    $u_j$ is added to set $S_2$
                **end if**
            **end for**
        **end if**
        **while** Not all node is included in $L$ **do**
            **if** Set $S_1$ is not empty **then**
                We sort the set $S_1$ by the sum of their distance to every free variable in $\mathcal{G}$, choose the most remote one, and if there is a tie, randomly choose one node, $u_i$ to be the next to explore. We remove $u_i$ from set $S_1$.
            **else**
                In this case, we know set $S_2$ is not empty because of the connectivity of $\mathcal{G}$. We randomly choose a node $u_i \in S_2$ to be the next node to explore. We remove $u_i$ from set $S_2$.
            **end if**
            **for** Node $u_j$ that connects to $u_i$ **do**
                **if** $f(u_j)$ is existential variable **then**
                    $u_j$ is added to set $S_1$
                **else**
                    $u_j$ is added to set $S_2$
                **end if**
            **end for**
            List $L$ append the node $u_i$
        **end while**
    **end for**
    **return** The list $L$ as the ordering of nodes in the whole abstract query graph $\mathcal{G}$
***

For CQD [2] and LMPNN [35], their original implementation does not require the operator tree, so we just use their original implementation. Specifically, in a query graph with multiple free variables, for CQD we predict the answer for each free variable individually as taking others free variables as existential variables, for LMPNN, we just got all embedding of nodes that represent free variables.

For FIT [39], though it is proposed to solve $EFO_1$ queries, it is computationally costly: it has a complexity of $O(\mathcal{E}^2)$ in the acyclic graphs and is even not polynomial in the cyclic graphs, the reason is that FIT degrades to enumeration to deal with cyclic graph. In our implementation, we further restrict FIT to at most enumerate 10 possible candidates for each node in the query graph, this practice has allowed FIT to be implemented in the dataset FB15k-237 [32]. However, it cost 20 hours to evaluate FIT on our $EFO_k$-CQA dataset while other models only need no more than two hours. Moreover, for larger knowledge graph, including NELL [7] and FB15k [5], we have also encountered an out-of-memory error in a Tesla V100 GPU with 32G memory when implementing FIT, thus, we omit its result in these two knowledge graphs.

# G   Further outlook to more complex query answering

In this section, we discuss possible further development in the task of complex query answering and how our work, especially our framework proposed in Section 4 can help with future development.

We list some new features that may be of interest and show the maximum versatility our framework can reach. Our analysis and characterization of future queries inherit the outlook in [37] and also is based on the current development.

**Inductive Reasoning** Inductive reasoning is a new trend in the field of complex query answering. Some entities [9] or even relations [15] are not seen in the training period, namely they can not be found by the observed knowledge graph $\mathcal{G}_o$ therefore, the inductive generalization is essential for the model to infer answers. We note that our framework is powerful enough to sample inductive queries with the observed knowledge graph $\mathcal{G}_o$ given. Therefore, the functionality of sampling inductive query is already contained and implemented in our framework, see `https://anonymous.4open.science/r/EFOK-CQA/README.md`.

**N-ary relation** N-ary relation is a relation that has $n > 2$ corresponding entities, therefore, the factual information in the knowledge graph is not a triple but a $(n + 1)$-tuple. Moreover, the query graph is also a hypergraph, making the corresponding CSP problem even harder. This is a newly introduced topic [23, 1] in complex query answering, which our framework has limitations in representing.

**Knowledge graph with attribute** Currently, there has been some research that has taken the additional attribute of the knowledge graph into account. Typical attributes include entity types [14], numerical literals [4],triple timestamps [16, 30], and triple probabilities [7]. We note that attributes expand the entity set $\mathcal{E}$ from all entities to entities with attribute values, it is also possible that the relation set $\mathcal{R}$ is also extended to contain corresponding relations, like "greater", "less" when dealing with numerical literals. Then, our framework can represent queries on such extended knowledge graphs like in [4], where no function like "plus", or "minus" is considered and the predicates are also binary.

Overall, our framework can be applied to some avant-garde problem settings given certain properties, thus those functionalities proposed in Section 4 can be useful. We hope our discussion helps with the future development of complex query answering.

# H  Additional experiment result and analysis

In this section, we offer another experiment result not available to be shown in the main paper. For the purpose of supplementation, we select some representative experiment result as the experiment result is extremely complex to be categorized and be shown. we present the further benchmark result of the following: the analysis of benchmark result in detail, more than just the averaged score in Table 1 and Table 2, which is provided in Appendix H.1; result of different knowledge graphs, including NELL and FB15k, which is provided in Appendix H.2 and H.3, the situation of more constant entities since we only discuss when there are two constant entities in Table 2, the result is provided in Appendix H.4, and finally, all queries(including the queries without marginal hard answers), in Appendix H.5.

We note that we have explained in Section 4.5 and Appendix E that for a query with multiple free variables, some or all of the free variables may not have their marginal hard answer and thus the marginal metric can not be computed. Therefore, in the result shown in Table 2 in Section 5.3, we only conduct evaluation on those queries that both of their free variables have marginal hard answers, and we offer the benchmark result of all queries in Appendix H.5 where only two kinds of metrics are available.

## H.1  Further result and analysis of the experiment in main paper

To supplement the experiment result already shown in Section 5.2 and Section 5.3, we have included more benchmark results in this section. Though the averaged score is a broadly-used statistic to benchmark the model performance on our $\text{EFO}_k$ queries, this is not enough and we have offered much more detail in this section.

**Whole combinatorial space helps to develop trustworthy machine learning models.** Firstly, we show more detailed benchmark results of the relative performance between our selected six CQA

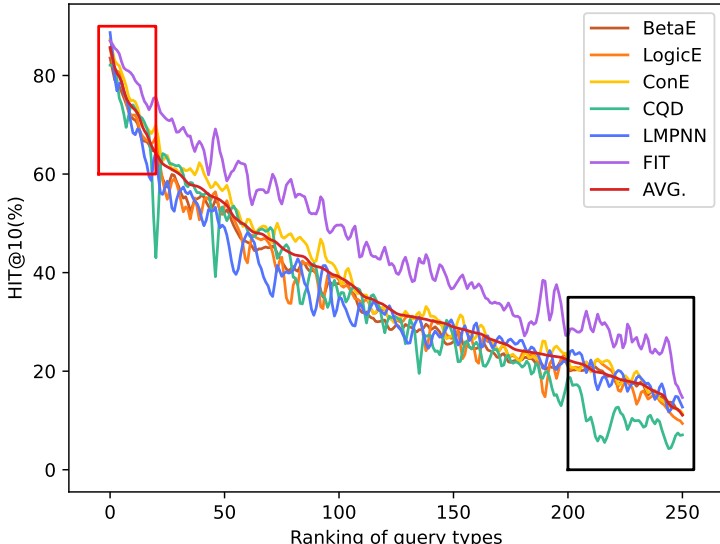

Figure 6: Relative performance of the six representative CQA models in referring queries with one free variable, where the ranking of query types is determined by the average HIT@10 score. A Gaussian filter with sigma=1 is added to smooth the curve. We also use the red box to highlight the easiest queries and the black box to highlight the most challenging ones.

models, the result is shown in Table 6. Specifically, we plot two boxes, the black one, including the most difficult query types, and the red box, including the easiest query types. In the easiest part, we find that even the worst model and the best model have pretty similar performance despite that they may differ greatly in other query types. The performance in the most difficult query types is more important when the users are risk-sensitive and desire a trustworthy machine-learning model that does not crash in extreme cases [33] and we highlight it in the black box. In the black box, we note that CQD [2], though designed in a rather general form, is pretty unstable when comes to empirical evaluation, as it has a clear downward curve and deviates from other model's performance enormously in the most difficult query types. Therefore, though its performance is better than LMPNN and comparable to BetaE on average as reported in 1, its unsteady performance suggests its inherent weakness. On the other hand, ConE [40] is much more steady and outperforms BetaE and LogicE consistently. We also show the result when there are two free variables in Figure 7, where the model performance is much less steady but the trend is similar to the $EFO_1$ case in general.

**Empirical hardness of query types and incomplete discussion of the previous dataset.** Moreover, we also discuss the empirical hardness of query types themselves and compare different datasets accordingly in Figure 8. We find the standard deviation of the six representative CQA models increases in the most difficult part and decreases in the easiest part, corroborating our discussion in the first paragraph. We also highlight those query types that have already been investigated in BetaE dataset [28] and FIT dataset [39]. We intuitively find that the BetaE dataset does not include very challenging query types while the FIT dataset mainly focuses on them. This can be explained by the fact that nine out of ten most challenging query types correspond to multigraph, which the BetaE dataset totally ignores while the FIT dataset highlights it as a key feature. To give a quantitative analysis of whether their hand-crafted query types are sampled from the whole combinatorial space, we have adopted the Kolmogorov–Smirnov test to test the distribution discrepancy between their distribution and the query type distribution in $EFO_k$-CQA since $EFO_k$-CQA enumerates all possible query types in the given combinatorial space and is thus unbiased. We find that the BetaE dataset is indeed generally easier and its p-value is 0.78, meaning that it has a 78 percent possibility to be

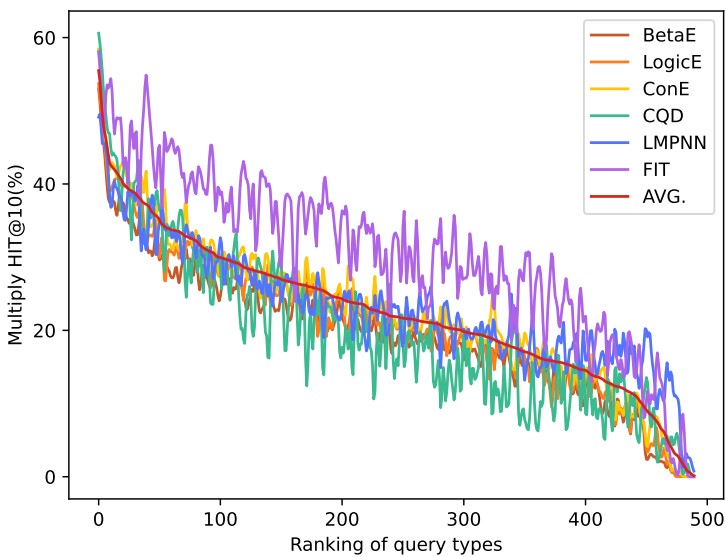

Figure 7: Relative performance of the six representative CQA models in referring queries with two free variables, the ranking of query types is determined by the average Multiply HIT@10 score. A Gaussian filter with sigma=1 is added to smooth the curve.

unbiased, while the FIT dataset is significantly harder and its p-value is 0.27. Therefore, there is no significant statistical evidence to prove they are sampled from the whole combinatorial space unbiasedly.

## H.2 Further benchmark result of $k$=1

Firstly, we present the benchmark result when there is only one free variable, since the result in FB15k-237 is provided in Table 1, we provide the result for other standard knowledge graphs, FB15k and NELL, their result is shown in Table 6 and Table 7, correspondingly. We note that FIT is out of memory with the two large graphs FB15k and NELL as explained in Appendix F and we do not include its result. As FB15k and NELL are both reported to be easier than FB15k-237, the models have better performance. The trend and analysis are generally similar to our discussion in Section 5.2 with some minor, unimportant changes that LogicE [24] has outperformed ConE [40] in the knowledge graph NELL, indicating one model may not perform identically well in all knowledge graphs.

## H.3 Further benchmark result for $k$=2 in more knowledge graphs

Then, similar to Section 5.3, we provide the result for other standard knowledge graphs, FB15k and NELL, when the number of constant entities is fixed to two, their result is shown in Table 8 and Table 9, correspondingly.

We note that though in some breakdowns, the marginal score is over 90 percent, almost close to 100 percent, the joint score is pretty slow, which further corroborates our findings that joint metric is significantly harder and more challenging in Section 5.3.

## H.4 Further benchmark result for $k$=2 with more constant numbers.

As the experiment in Section 5.3 only contains the situation where the number of constant entity is fixed as one, we offer the further experiment result in Table 10.

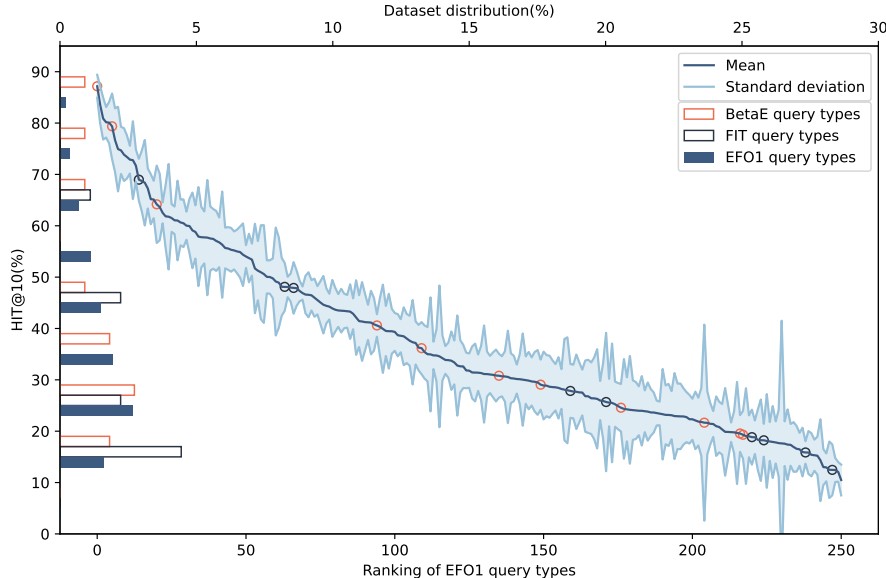

Figure 8: Query type distribution in three different datasets, BetaE one, FIT one, and the $EFO_1$ part in our $EFO_k$-CQA dataset. The left part shows the histogram that represents the probability density function of each dataset. The ranking of query types is also determined by the mean HIT@10 score as in Figure 6, with the standard deviation of the performance of the six CQA models shown as the light blue error bar.

The result shows that models perform worse with fewer constant variables when compares to the result in Table 2, this observation is the same as the previous result with one free variable that has been discussed in Section 5.2.

## H.5 Further benchmark result for $k$=2 including all queries

Finally, as we have explained in Section 4.5 and Appendix E, there are some valid $EFO_k$ queries without marginal hard answers when $k > 1$. Thus, there is no way to calculate the marginal scores, all our previous experiments are therefore only conducted on those queries that all their free variables have marginal hard answers. In this section, we only present the result of the Multiply and Joint score, as they can be computed for any valid $EFO_k$ queries, and therefore this experiment is conducted on the whole $EFO_k$-CQA dataset.

We follow the practice in Section 5.3 that fixed the number of constant entities as two, as the impact of constant entities is pretty clear, which has been further corroborated in Appendix H.4. The experiments are conducted on all three knowledge graphs, FB15k-237, FB15k, and NELL, the result is shown in Table 11, Table 12, and Table 13, correspondingly.

Interestingly, comparing the result in Table 2 and Table 11, the multiple scores actually increase through the joint scores are similar. This may be explained by the fact that if one free variable has no marginal hard answer, then it can be easily predicted, leading to a better performance for the whole query.

Table 5: MRR scores(%) for inferring queries with one free variable on FB15k-237. We denote $e$ as the number of existential variables and $c$ as the number of constant entities. SDAG represents the Simple Directed Acyclic Graph, Multi for multigraph, and Cyclic for the cyclic graph. AVG.($c$) and AVG.($e$) is the average score of queries with the number of constant entities / existential variables fixed.

| Model | $e$ / $c$ | 0 | 1 | | 2 | | | AVG.($c$) | AVG. |
|---|---|---|---|---|---|---|---|---|---|
| | | SDAG | SDAG | Multi | SDAG | Multi | Cyclic | | |
| BetaE | 1 | 16.2 | 17.9 | 10.9 | 10.6 | 8.5 | 16.5 | 11.1 | |
| | 2 | 35.6 | 20.2 | 19.1 | 15.7 | 15.7 | 27.1 | 17.8 | 20.7 |
| | 3 | 53.3 | 32.4 | 33.1 | 21.7 | 21.6 | 37.4 | 24.8 | |
| | AVG.($e$) | 37.4 | 25.7 | 23.5 | 18.8 | 18.1 | 30.5 | | |
| LogicE | 1 | 17.4 | 19.0 | 11.5 | 11.0 | 8.5 | 16.8 | 11.5 | |
| | 2 | 36.7 | 21.2 | 19.8 | 16.5 | 16.1 | 27.3 | 18.4 | 21.3 |
| | 3 | 55.5 | 34.6 | 34.5 | 22.3 | 22.0 | 37.5 | 25.4 | |
| | AVG.($e$) | 38.9 | 27.3 | 24.5 | 19.4 | 18.5 | 30.6 | | |
| ConE | 1 | 18.6 | 19.9 | 11.8 | 11.4 | 9.3 | 18.7 | 12.3 | |
| | 2 | 39.1 | 22.4 | 20.8 | 18.1 | 17.6 | 30.7 | 20.1 | 23.1 |
| | 3 | 58.8 | 36.4 | 37.0 | 24.6 | 23.8 | 41.7 | 27.6 | |
| | AVG.($e$) | 41.4 | 28.7 | 26.0 | 21.3 | 20.1 | 34.2 | | |
| CQD | 1 | **22.2** | 19.5 | 9.0 | 9.2 | 6.4 | 15.6 | 10.0 | |
| | 2 | 35.3 | 20.1 | 19.1 | 16.4 | 16.2 | 27.6 | 18.4 | 21.9 |
| | 3 | 40.3 | 32.9 | 34.3 | 24.4 | 24.0 | 40.2 | 26.8 | |
| | AVG.($e$) | 33.9 | 26.2 | 23.7 | 20.5 | 19.4 | 31.9 | | |
| LMPNN | 1 | 20.5 | 21.4 | 11.2 | 11.6 | 8.7 | 17.0 | 11.9 | |
| | 2 | 42.0 | 22.6 | 18.5 | 16.5 | 14.9 | 26.5 | 17.9 | 20.5 |
| | 3 | 62.3 | 35.9 | 31.6 | 22.1 | 19.8 | 35.5 | 24.0 | |
| | AVG.($e$) | 44.2 | 28.8 | 22.7 | 19.4 | 16.9 | 29.4 | | |
| FIT | 1 | **22.2** | **25.0** | **17.4** | **13.9** | **11.7** | **23.3** | **15.6** | |
| | 2 | **45.3** | **29.6** | **28.5** | **23.8** | **24.3** | **35.5** | **26.5** | **30.3** |
| | 3 | **64.5** | **44.8** | **45.4** | **33.3** | **33.5** | **44.4** | **36.2** | |
| | AVG.($e$) | **46.7** | **36.2** | **33.6** | **28.6** | **27.9** | **37.9** | | |

Table 6: MRR scores(%) for inferring queries with one free variable on FB15k. The notation of $e$, $c$, SDAG, Multi, Cyclic, AVG.($c$) and AVG.($e$) are the same as Table 1.

| Model | $c$ \ $e$ | 0 | 1 | | 2 | | | AVG.($c$) | AVG. |
|---|---|---|---|---|---|---|---|---|---|
| | | SDAG | SDAG | Multi | SDAG | Multi | Cyclic | | |
| BetaE | 1 | 38.6 | 30.4 | 29.2 | 21.7 | 21.7 | 24.1 | 24.3 | |
| | 2 | 49.7 | 34.0 | 37.2 | 28.3 | 29.2 | 35.5 | 31.0 | 34.0 |
| | 3 | 63.5 | 46.4 | 48.6 | 33.9 | 36.1 | 45.8 | 38.1 | |
| | AVG.($e$) | 63.5 | 46.4 | 48.6 | 33.9 | 36.1 | 45.8 | 38.1 | |
| LogicE | 1 | 46.0 | 33.8 | 32.1 | 23.3 | 22.8 | 25.6 | 26.2 | |
| | 2 | 51.2 | 35.9 | 39.0 | 30.6 | 30.5 | 36.9 | 32.7 | 35.6 |
| | 3 | 64.5 | 48.6 | 49.8 | 35.4 | 37.5 | 47.7 | 39.6 | |
| | AVG.($e$) | 54.9 | 41.7 | 42.3 | 32.8 | 33.4 | 40.4 | | |
| ConE | 1 | 52.5 | 35.8 | 34.9 | 25.9 | 25.9 | 29.5 | 29.3 | |
| | 2 | 57.0 | 40.0 | 43.4 | 33.2 | 34.2 | 40.8 | 36.3 | 39.5 |
| | 3 | 70.6 | 53.1 | 55.3 | 39.3 | 41.8 | 52.5 | 43.9 | |
| | AVG.($e$) | 61.0 | 45.6 | 46.8 | 36.1 | 37.4 | 44.8 | | |
| CQD | 1 | 74.6 | 36.1 | 32.7 | 17.6 | 16.7 | 25.4 | 23.7 | |
| | 2 | 52.2 | 35.2 | 40.9 | 29.2 | 31.5 | 39.2 | 33.2 | 37.2 |
| | 3 | 53.3 | 32.4 | 33.1 | 21.7 | 21.6 | 37.4 | 24.8 | |
| | AVG.($e$) | 59.4 | 41.5 | 44.6 | 33.3 | 35.3 | 43.3 | | |
| LMPNN | 1 | 63.7 | 39.9 | 35.3 | 28.7 | 26.4 | 28.7 | 30.7 | |
| | 2 | 65.0 | 41.9 | 38.8 | 34.4 | 31.7 | 38.4 | 35.1 | 37.7 |
| | 3 | 79.8 | 54.0 | 49.5 | 38.9 | 37.1 | 48.0 | 40.8 | |
| | AVG.($e$) | 70.2 | 47.4 | 42.8 | 36.6 | 34.1 | 41.6 | | |

Table 7: MRR scores(%) for inferring queries with one free variable on NELL. The notation of $e$, $c$, SDAG, Multi, Cyclic, AVG.($c$) and AVG.($e$) are the same as Table 1.

| Model | $c$ \ $e$ | 0 | 1 | | 2 | | | AVG.($c$) | AVG. |
|---|---|---|---|---|---|---|---|---|---|
| | | SDAG | SDAG | Multi | SDAG | Multi | Cyclic | | |
| BetaE | 1 | 13.9 | 26.4 | 35.0 | 8.6 | 14.9 | 19.1 | 17.5 | |
| | 2 | 58.8 | 31.5 | 43.8 | 22.4 | 30.6 | 34.7 | 30.7 | 33.6 |
| | 3 | 78.8 | 48.6 | 58.3 | 29.6 | 39.0 | 47.0 | 39.5 | |
| | AVG.($e$) | 53.1 | 38.5 | 48.3 | 25.2 | 33.3 | 38.2 | | |
| LogicE | 1 | 18.3 | 29.2 | 39.6 | 12.1 | 19.0 | 20.4 | 21.1 | |
| | 2 | 63.5 | 34.4 | 47.3 | 26.4 | 34.0 | 37.6 | 34.2 | 36.9 |
| | 3 | 79.6 | 51.2 | 59.3 | 33.1 | 42.2 | 50.1 | 42.6 | |
| | AVG.($e$) | 56.3 | 41.3 | 50.9 | 28.8 | 36.7 | 41.0 | | |
| ConE | 1 | 16.7 | 26.9 | 36.6 | 11.1 | 16.9 | 22.3 | 19.6 | |
| | 2 | 60.5 | 33.6 | 46.6 | 25.3 | 33.1 | 40.1 | 33.6 | 36.6 |
| | 3 | 79.9 | 50.6 | 59.2 | 33.2 | 42.2 | 52.6 | 42.8 | |
| | AVG.($e$) | 54.9 | 40.3 | 50.0 | 28.4 | 36.2 | 43.4 | | |
| CQD | 1 | 22.3 | 30.6 | 37.3 | 13.3 | 17.9 | 20.7 | 20.9 | |
| | 2 | 59.8 | 34.0 | 45.2 | 28.8 | 35.4 | 38.9 | 35.3 | 38.2 |
| | 3 | 62.7 | 48.8 | 59.9 | 36.4 | 44.1 | 52.6 | 44.3 | |
| | AVG.($e$) | 50.1 | 40.2 | 49.9 | 31.6 | 38.1 | 42.7 | | |
| LMPNN | 1 | 20.7 | 29.8 | 33.3 | 13.4 | 16.5 | 21.8 | 19.8 | |
| | 2 | 63.5 | 35.4 | 43.3 | 27.0 | 30.2 | 37.6 | 32.3 | 35.1 |
| | 3 | 80.8 | 50.7 | 56.0 | 33.6 | 39.2 | 47.6 | 40.7 | |
| | AVG.($e$) | 57.4 | 41.5 | 46.7 | 29.4 | 33.6 | 40.0 | | |

Table 8: HIT@10 scores(%) of three different types for answering queries with two free variables on FB15k. The constant number is fixed to be two. The notation of $e$, SDAG, Multi, and Cyclic is the same as Table 2.

| Model | HIT@10 Type | $e = 0$ | | $e = 1$ | | | $e = 2$ | | | AVG. |
|---|---|---|---|---|---|---|---|---|---|---|
| | | SDAG | Multi | SDAG | Multi | Cyclic | SDAG | Multi | Cyclic | |
| BetaE | Marginal | 76.9 | 77.2 | 68.9 | 69.3 | 75.1 | 55.0 | 57.4 | 73.6 | 63.6 |
| | Multiply | 41.7 | 41.6 | 31.7 | 31.0 | 38.7 | 25.2 | 25.9 | 36.1 | 29.7 |
| | Joint | 11.6 | 13.7 | 8.7 | 8.6 | 17.8 | 4.9 | 5.4 | 14.3 | 8.4 |
| LogicE | Marginal | 82.9 | 80.9 | 73.6 | 72.9 | 76.6 | 58.9 | 60.7 | 75.7 | 66.9 |
| | Multiply | 47.5 | 45.0 | 36.3 | 34.1 | 40.4 | 28.5 | 29.0 | 38.0 | 32.7 |
| | Joint | 12.7 | 13.9 | 10.0 | 9.9 | 19.2 | 6.1 | 6.5 | 15.9 | 9.6 |
| ConE | Marginal | 84.1 | 84.8 | 76.5 | 76.3 | 81.4 | 61.8 | 63.8 | 79.7 | 70.2 |
| | Multiply | 48.7 | 48.1 | 37.7 | 35.9 | 44.2 | 29.9 | 30.4 | 41.4 | 34.6 |
| | Joint | 14.2 | 15.6 | 10.3 | 10.4 | 20.6 | 6.2 | 6.6 | 16.9 | 10.1 |
| CQD | Marginal | 73.8 | 76.8 | 69.0 | 71.9 | 76.3 | 51.1 | 54.4 | 77.0 | 62.9 |
| | Multiply | 45.0 | 46.6 | 37.4 | 36.9 | 43.9 | 28.1 | 29.2 | 41.9 | 34.0 |
| | Joint | 17.1 | 19.0 | 13.1 | 13.0 | 20.6 | 7.7 | 8.6 | 18.1 | 11.9 |
| LMPNN | Marginal | 89.2 | 80.1 | 80.3 | 78.2 | 84.2 | 65.6 | 63.7 | 80.2 | 71.3 |
| | Multiply | 56.6 | 50.5 | 45.7 | 42.4 | 49.0 | 37.6 | 34.8 | 44.6 | 39.7 |
| | Joint | 18.9 | 17.2 | 12.9 | 12.4 | 22.4 | 8.0 | 7.5 | 16.9 | 11.2 |

Table 9: HIT@10 scores(%) of three different types for answering queries with two free variables on NELL. The constant number is fixed to be two. The notation of $e$, SDAG, Multi, and Cyclic is the same as Table 2.

| Model | HIT@10 Type | $e = 0$ | | $e = 1$ | | | $e = 2$ | | | AVG. |
|---|---|---|---|---|---|---|---|---|---|---|
| | | SDAG | Multi | SDAG | Multi | Cyclic | SDAG | Multi | Cyclic | |
| BetaE | Marginal | 81.3 | 95.9 | 72.8 | 85.5 | 79.9 | 57.2 | 66.7 | 77.0 | 71.2 |
| | Multiply | 48.2 | 56.7 | 41.3 | 46.1 | 47.6 | 33.1 | 36.5 | 42.9 | 39.6 |
| | Joint | 19.2 | 31.8 | 21.2 | 26.5 | 21.7 | 13.8 | 17.5 | 18.5 | 18.8 |
| LogicE | Marginal | 87.1 | 99.8 | 81.0 | 91.8 | 83.2 | 65.7 | 74.0 | 81.0 | 77.7 |
| | Multiply | 52.5 | 60.3 | 47.6 | 51.7 | 50.2 | 39.4 | 42.6 | 46.0 | 44.8 |
| | Joint | 21.1 | 32.8 | 25.4 | 30.5 | 23.3 | 18.0 | 21.5 | 20.5 | 22.3 |
| ConE | Marginal | 82.6 | 96.4 | 76.0 | 87.8 | 88.1 | 60.0 | 69.3 | 83.0 | 74.7 |
| | Multiply | 48.7 | 56.9 | 41.9 | 46.3 | 52.2 | 34.5 | 38.1 | 47.7 | 41.7 |
| | Joint | 17.0 | 30.9 | 19.3 | 25.0 | 24.9 | 12.9 | 17.2 | 20.3 | 18.8 |
| CQD | Marginal | 79.5 | 96.3 | 83.2 | 92.2 | 83.5 | 65.8 | 75.7 | 84.8 | 79.4 |
| | Multiply | 49.2 | 57.8 | 51.1 | 53.1 | 51.4 | 40.6 | 45.1 | 50.6 | 47.4 |
| | Joint | 23.0 | 38.0 | 29.7 | 34.2 | 26.4 | 21.4 | 25.4 | 24.0 | 26.0 |
| LMPNN | Marginal | 88.5 | 96.6 | 81.5 | 90.9 | 85.3 | 65.0 | 70.7 | 83.1 | 76.7 |
| | Multiply | 55.7 | 62.4 | 50.3 | 53.3 | 54.0 | 40.8 | 42.6 | 50.3 | 46.5 |
| | Joint | 23.4 | 36.4 | 25.5 | 29.4 | 24.0 | 16.6 | 19.7 | 21.5 | 21.5 |

Table 10: HIT@10 scores(%) of three different types for answering queries with two free variables on FB15k-237. The constant number is fixed to be one. The notation of $e$, SDAG, Multi, and Cyclic is the same as Table 2.

| Model | HIT@10 Type | $e = 0$ | | $e = 1$ | | | $e = 2$ | | | AVG. |
|---|---|---|---|---|---|---|---|---|---|---|
| | | SDAG | Multi | SDAG | Multi | Cyclic | SDAG | Multi | Cyclic | |
| BetaE | Marginal | 37.5 | 29.7 | 33.4 | 28.1 | 35.6 | 30.0 | 25.9 | 41.2 | 31.2 |
| | Multiply | 18.9 | 13.7 | 15.3 | 10.3 | 15.2 | 17.7 | 13.3 | 17.2 | 14.3 |
| | Joint | 0.9 | 1.1 | 1.4 | 0.9 | 3.3 | 1.1 | 0.9 | 3.9 | 1.7 |
| LogicE | Marginal | 40.6 | 30.7 | 36.0 | 29.1 | 34.6 | 29.8 | 25.3 | 41.5 | 31.4 |
| | Multiply | 21.1 | 14.3 | 17.2 | 10.9 | 16.3 | 17.8 | 13.3 | 17.5 | 14.7 |
| | Joint | 1.4 | 1.4 | 1.6 | 0.9 | 3.7 | 1.4 | 1.0 | 4.3 | 1.9 |
| ConE | Marginal | 40.8 | 32.4 | 37.3 | 30.4 | 40.7 | 31.1 | 26.9 | 45.0 | 33.5 |
| | Multiply | 22.1 | 15.2 | 18.4 | 11.7 | 19.3 | 18.5 | 14.8 | 20.9 | 16.5 |
| | Joint | 1.4 | 1.0 | 1.7 | 1.0 | 4.3 | 1.4 | 1.0 | 4.4 | 2.0 |
| CQD | Marginal | 73.8 | 76.8 | 69.0 | 71.9 | 76.3 | 51.1 | 54.4 | 77.0 | 62.9 |
| | Multiply | 23.3 | 9.1 | 18.5 | 9.2 | 16.2 | 14.6 | 9.2 | 19.1 | 12.9 |
| | Joint | 1.5 | 0.6 | 2.0 | 1.1 | 3.4 | 1.5 | 0.9 | 4.4 | 1.9 |
| LMPNN | Marginal | 39.0 | 27.6 | 40.0 | 29.5 | 39.3 | 30.6 | 24.8 | 42.7 | 32.0 |
| | Multiply | 25.1 | 13.9 | 24.3 | 13.3 | 21.6 | 20.0 | 14.0 | 21.1 | 17.1 |
| | Joint | 1.6 | 1.3 | 2.5 | 1.3 | 3.9 | 1.5 | 1.0 | 4.0 | 2.0 |

Table 11: HIT@10 scores(%) of two different types for answering queries with two free variables on FB15k-237(including queries without the marginal hard answer). The constant number is fixed to be two. The notation of $e$, SDAG, Multi, and Cyclic is the same as Table 2.

| Model | HIT@10 Type | $e = 0$ | | $e = 1$ | | | $e = 2$ | | | AVG. |
|---|---|---|---|---|---|---|---|---|---|---|
| | | SDAG | Multi | SDAG | Multi | Cyclic | SDAG | Multi | Cyclic | |
| BetaE | Multiply | 29.1 | 29.1 | 18.3 | 37.5 | 10.4 | 28.0 | 93.6 | 74.6 | 24.1 |
| | Joint | 2.1 | 2.2 | 1.7 | 3.0 | 2.4 | 1.8 | 5.8 | 14.2 | 4.6 |
| LogicE | Multiply | 31.6 | 32.9 | 19.8 | 39.6 | 10.9 | 28.7 | 96.3 | 73.8 | 25.4 |
| | Joint | 2.6 | 2.5 | 2.1 | 3.1 | 2.5 | 2.2 | 6.4 | 15.6 | 5.0 |
| ConE | Multiply | 32.6 | 31.9 | 20.5 | 41.0 | 12.6 | 29.0 | 99.7 | 86.8 | 27.0 |
| | Joint | 3.0 | 2.1 | 1.9 | 3.3 | 2.7 | 2.2 | 6.6 | 16.8 | 5.4 |
| CQD | Multiply | 34.5 | 23.4 | 22.3 | 36.8 | 10.6 | 26.4 | 75.3 | 77.3 | 25.6 |
| | Joint | 2.9 | 1.4 | 2.1 | 3.3 | 2.3 | 2.0 | 5.0 | 15.0 | 5.6 |
| LMPNN | Multiply | 36.8 | 29.3 | 27.5 | 45.8 | 13.9 | 31.2 | 97.0 | 86.5 | 27.9 |
| | Joint | 2.7 | 2.2 | 2.7 | 3.9 | 2.5 | 2.1 | 5.8 | 14.6 | 5.0 |
| FIT | Multiply | 41.5 | 44.4 | 28.9 | 56.8 | 10.2 | 39.4 | 139.7 | 100.3 | 35.0 |
| | Joint | 2.4 | 2.3 | 2.1 | 3.4 | 1.6 | 2.2 | 7.4 | 15.4 | 5.9 |

Table 12: HIT@10 scores(%) of two different types for answering queries with two free variables on FB15k(including queries without the marginal hard answer). The constant number is fixed to be two. The notation of $e$, SDAG, Multi, and Cyclic is the same as Table 2.

| Model | HIT@10 Type | $e = 0$ | | $e = 1$ | | | $e = 2$ | | | AVG. |
|---|---|---|---|---|---|---|---|---|---|---|
| | | SDAG | Multi | SDAG | Multi | Cyclic | SDAG | Multi | Cyclic | |
| BetaE | Multiply | 42.1 | 57.2 | 26.5 | 66.5 | 15.5 | 34.6 | 134.9 | 100.0 | 35.0 |
| | Joint | 6.6 | 9.4 | 4.5 | 10.2 | 4.6 | 4.3 | 16.7 | 26.0 | 9.2 |
| LogicE | Multiply | 48.2 | 65.6 | 31.0 | 71.6 | 16.8 | 37.8 | 143.9 | 105.8 | 38.1 |
| | Joint | 7.5 | 11.2 | 5.6 | 12.5 | 5.3 | 5.6 | 20.4 | 28.5 | 10.5 |
| ConE | Multiply | 50.2 | 72.2 | 32.8 | 74.6 | 18.3 | 38.3 | 149.3 | 114.3 | 40.4 |
| | Joint | 6.8 | 10.0 | 5.2 | 12.5 | 5.5 | 5.2 | 19.4 | 30.4 | 11.0 |
| CQD | Multiply | 48.1 | 55.9 | 31.9 | 69.0 | 15.8 | 29.5 | 93.5 | 103.2 | 37.6 |
| | Joint | 9.4 | 11.4 | 6.6 | 14.8 | 4.8 | 5.5 | 17.5 | 27.2 | 12.0 |
| LMPNN | Multiply | 58.4 | 79.5 | 43.1 | 94.6 | 21.3 | 40.9 | 146.2 | 135.9 | 45.0 |
| | Joint | 8.6 | 12.9 | 6.8 | 15.6 | 6.2 | 5.4 | 19.3 | 31.7 | 11.6 |

Table 13: HIT@10 scores(%) of two different types for answering queries with two free variables on NELL(including queries without the marginal hard answer). The constant number is fixed to be two. The notation of $e$, SDAG, Multi, and Cyclic is the same as Table 2.

| Model | HIT@10 Type | $e = 0$ | | $e = 1$ | | | $e = 2$ | | | AVG. |
|---|---|---|---|---|---|---|---|---|---|---|
| | | SDAG | Multi | SDAG | Multi | Cyclic | SDAG | Multi | Cyclic | |
| BetaE | Multiply | 21.2 | 47.3 | 22.0 | 51.9 | 14.7 | 24.1 | 80.5 | 79.7 | 33.4 |
| | Joint | 4.2 | 19.6 | 6.8 | 19.1 | 5.1 | 6.8 | 26.7 | 24.0 | 14.1 |
| LogicE | Multiply | 26.6 | 52.8 | 28.8 | 63.4 | 16.0 | 32.8 | 103.1 | 88.5 | 38.9 |
| | Joint | 3.8 | 21.5 | 9.7 | 26.0 | 5.9 | 11.5 | 36.9 | 27.3 | 16.5 |
| ConE | Multiply | 25.3 | 51.4 | 23.9 | 53.9 | 16.9 | 27.3 | 90.7 | 90.6 | 36.7 |
| | Joint | 3.4 | 20.2 | 6.4 | 17.0 | 6.1 | 7.2 | 27.0 | 27.1 | 14.2 |
| CQD | Multiply | 30.3 | 48.9 | 30.6 | 64.3 | 15.9 | 33.1 | 88.9 | 91.2 | 40.9 |
| | Joint | 4.4 | 21.9 | 9.8 | 27.5 | 5.6 | 12.0 | 37.6 | 28.1 | 18.0 |
| LMPNN | Multiply | 33.4 | 58.3 | 33.7 | 65.3 | 19.4 | 30.7 | 85.1 | 105.0 | 41.8 |
| | Joint | 4.4 | 23.7 | 10.0 | 21.9 | 5.8 | 8.2 | 23.2 | 28.8 | 15.7 |