# OpenReview forum: "${\rm EFO}_k$-CQA: Towards Knowledge Graph Complex Query Answering beyond Set Operation"
_NeurIPS.cc/2023/Track/Datasets_and_Benchmarks — Submitted to NeurIPS 2023 Datasets and Benchmarks_

### Official Review · Reviewer_t4bs · 2023-07-17

**Rating:** 6
**Confidence:** 5
**Correctness:** OK
**Clarity:** OK

**Strengths:**

S1. The proposed dataset adds more query patterns to the standard suite of 14 standard patterns used in the query answering literature.
S2. The authors describe the methodology of the generation process and motivation behind the evaluation metrics.


**Additional Feedback:**

It would be more clear to say in captions to Tables 1 and 2 that the tables report query performance on queries with one and two **free** variables (answer nodes), respectively.

**Documentation:**

The supplementary includes the codebase with some readme files and comments.

**Ethics:**

OK

**Limitations:**

The authors did not discuss limitations in the main text nor in the appendices. “We can not handle queries with the universal quantifier” from the checklist is not really a point worth discussing (the literature omits universal quantifiers and focuses on other areas). I’d expect to see a discussion on the limitations stemming from W1 and W2.


**Opportunities For Improvement:**

**W1.** The main drawback of the work is the focus on transductive, conjunctive, entity-centric queries on triple-based graphs, a rather stale setup. Extending the previous work, the EFO-1 dataset, the authors introduce EFO-k with $k$ existentially quantified variables with even more query patterns, but the crux of the problem remains the same. In light of the growing interest towards inductive reasoning, queries with numeric literals, and more graph modalities, the classic transductive, entity-centric triple-based queries seem to be “beaten to death”. That is, what kind of additional useful information do we get from evaluating query answering models on 700+ new patterns that we can’t get from, eg, 100 patterns? As it is getting hard (or barely possible) to analyze performance and fail cases in each particular pattern out of 700+, do those hundreds of patterns carry any value if we anyways report averaged performance? Wouldn’t it be more feasible to have a representative sample of 5-10% of query patterns that would reliably estimate the overall performance? I’d hypothesize that the reason why EFO-1 (the dataset EFO-k is largely based on) hasn’t been used that much in the literature is the already daunting amount of query patterns to evaluate on.

**W2.** The limitations imposed on query patterns (eg, the maximum number of edges is not more than the number of nodes in the query graph, so no multi-edge and cyclic queries at the same time) look rather artificial and motivated by the chosen benchmarked systems that _can_  answer certain query types. I believe the value of new datasets and benchmarks is also catering for the cases that existing systems might _not_ handle (or handle poorly) such that the dataset remains challenging in the next few years of research in this area. The EFO-k queries can be solved by many existing query answering methods.


**Relation To Prior Work:**

OK

**Summary And Contributions:**

The paper focuses on the complex logical query answering domain where the task is to predict answers to logical queries (a collection of relational patterns organized into a query graph by logical operations like conjunction, disjunction, and negation) given an underlying incomplete knowledge graph. Traditionally, existing datasets include query patterns that assume only one leaf node in the query graph, that is, only one node denoting the answers. In this work, the authors build a new dataset for queries with more than one answer node and several possible existentially quantified variables.

---

> ### Author Response · Authors · 2023-08-21
>
> Thank you for your insightful comment. We value your constructive advice and have put much effort to improve our paper, which is detailed also in our general feedback and we welcome you to check that. We hope that our effort can address your issue.
>
>
> > The main drawback of the work is the focus on transductive, conjunctive, entity-centric queries on triple-based graphs, a rather stale setup.  Extending the previous work, the EFO-1 dataset, the authors introduce EFO-k with k existentially quantified variables with even more query patterns, but the crux of the problem remains the same. In light of the growing interest towards inductive reasoning, queries with numeric literals, and more graph modalities, the classic transductive, entity-centric triple-based queries seem to be “beaten to death”.
>
> Thank you for bringing up those recent research topics within the area of more complex query answering. We are also aware of these new developments, inductive reasoning models or models that can deal with numeric values.
>
> In fact, our framework proposed in Section 4 is **versatile enough** to generate queries in more complex settings. For example, we can sample inductive queries and we add the discussion in Appendix G, we hope you may check that.
>
> Moreover, we mainly focus on the traditional transductive task that is still prevalent today and serves as a foundation for more advanced settings[1], while those avant-garde settings have been put forward very recently and thus there are not many models that are capable to be benchmarked yet. Therefore, choosing the transductive setting helps us to focus more on  the dataset construction and empirical benchmark result. We also believe that extending the investigated query to EFOk itself is a milestone in the development of CQA.
>
> [1] Galkin, Michael, et al. "Inductive logical query answering in knowledge graphs." Advances in Neural Information Processing Systems 35 (2022): 15230-15243.
>
>
> > That is, what kind of additional useful information do we get from evaluating query answering models on 700+ new patterns that we can’t get from, eg, 100 patterns? As it is getting hard (or barely possible) to analyze performance and fail cases in each particular pattern out of 700+, do those hundreds of patterns carry any value if we anyways report averaged performance? Wouldn’t it be more feasible to have a representative sample of 5-10% of query patterns that would reliably estimate the overall performance?
>
> Thank you for mentioning the drawback of merely reporting the average score. Based on your feedback, we have now studied the overall distribution of the benchmark result over the 700+ query types and give new insight in Appendix H.1, where we discuss two issues that further help with the development of CQA models: how our EFOk-CQA dataset can help to detect untrustworthy machine learning models, and whether the current dataset is enough to benchmark the performance of current CQA models. We believe that these analyses can be only got by a thorough analysis of the whole combinatorial space.
>
>
> >  The limitations imposed on query patterns (eg, the maximum number of edges is not more than the number of nodes in the query graph, so no multi-edge and cyclic queries at the same time) look rather artificial and motivated by the chosen benchmarked systems that can answer certain query types. I believe the value of new datasets and benchmarks is also catering for the cases that existing systems might not handle (or handle poorly) such that the dataset remains challenging in the next few years of research in this area. The EFO-k queries can be solved by many existing query answering methods.
>
> The limitation is neither motivated by any specific models nor the evaluation protocol. In fact, we can explore **arbitrary combinatorial space** within the syntax of EFOk if necessary. The rationale behind “maximum number of edges is not more than the number of nodes” is to restrict the exponentially growing number of query types(namely abstract query graph), in fact, merely allowing the number of edges may exceed the number of nodes by one will create the total query type to **1539**. Moreover, this setting disentangles different features of the query graph and helps us to make more clear discussions about the impact of topology. Additionally, we believe our proposed EFOk query is very challenging from the current perspective, HIT@10 that we use in Table 2 is a rather easy metric, yet all current models are under 10% in the joint HIT@10, indicating they can barely handle those queries with multiple free variables.
>
> > It would be more clear to say in captions to Tables 1 and 2 that the tables report query performance on queries with one and two free variables (answer nodes), respectively.
>
> We have correctified it, thank you for your advice.

---

> > ### Author Response · Authors · 2023-08-29
> > **Looking for reply**
> >
> > Dear reviewer:
> >
> > We thank you for your review and suggestions. We have made significant modifications to our paper accordingly, adding a large number of new discussions and analyses into our paper (most of them have to be included in the Appendix because of the space limit, and the new material is highlighted in blue color).
> >
> > We hope that our improvement of the paper can address many of your concerns raised in your review and help our paper be understood better, and we are looking forward to your reply.
> >
> > Best regards,
> >
> > Paper346 Authors

---

### Official Review · Reviewer_tAiM · 2023-07-20
**Review for Submission 346**

**Rating:** 6
**Confidence:** 3
**Correctness:** yes.
**Clarity:** Examples are included to enhance the …

**Strengths:**

1. A dataset with 741 query types is constructed.
2. Experiments have been conducted on the newly constructed dataset.


**Additional Feedback:**

see above.

**Documentation:**

A URL to the code has been provided.

**Ethics:**

no.

**Limitations:**

see above

**Opportunities For Improvement:**

1.	It would be interesting and critical to give the percentage of nontrivial queries in real-world applications.

2.	The notation “V” in Section 2.1 is not given.


**Relation To Prior Work:**

Yes.

**Summary And Contributions:**

Answering complex queries over knowledge graphs is an interesting task. This paper proposes a new CQA dataset that covers the combinatorial space of EFO_k.

---

> ### Author Response · Authors · 2023-08-21
>
> Thank you for your comment. We have put much effort to improve our paper, which is detailed also in our general feedback and we welcome you to check that. We hope that our effort can address your issue.
>
>
> > It would be interesting and critical to give the percentage of nontrivial queries in real-world applications.
>
> Thank you for your advice though it is unclear what you mean by nontrivial queries in real-world applications. If you are concerned about the real-world application of CQA, we want to point out that the task of complex query answering has real-world applications, like fact ranking[1], and explainable recommendation[2]. However, some important practical applications can not be covered by existing dataset in complex query answering, because their construction is biased and have not discussed some key features of the query.
>
> We would like to introduce **one example in fraud detection** where we need to detect a group of people with cyclic money flow for anti-money laundering applications[3], we also note that this finding is also shared by open-source graph database[4,5]. Therefore, our investigation on cyclic queries and queries with more than one free variable can be justified to help develop more versatile CQA models that are suitable for more real-world applications.
>
> In comparison with the TPC-H which only has dozens of query types, we also believe that our thorough exploration of the whole combinatorial space (mentioned in Section 4.1 and carefully designed and explained in Appendix D.4) is important because our setting is fundamentally different than the TPC-H. The empirical result of a more complex query is hard to predict from other queries because of the open world assumption, as well as the fact that error can accumulate in the machine learning models, different from traditional database settings. Moreover, our thorough exploration indeed can help to detect untrustworthy machine learning models. We have included more discussions regarding this issue, we hope you may check the 5th part of our general rebuttal.
>
> > The notation “V” in Section 2.1 is not given.
>
> Thank you for your comment. This is a typo, the notation “V” should be $\mathcal{E}$, and we have corrected it.
>
>
> [1] Ren, Hongyu, et al. "Fact Ranking over Large-Scale Knowledge Graphs with Reasoning Embedding Models." Data Engineering: 124.
>
> [2] Syed, Muzamil Hussain, Tran Quoc Bao Huy, and Sun-Tae Chung. "Context-aware explainable recommendation based on domain knowledge graph." Big Data and Cognitive Computing 6.1 (2022): 11.
>
> [3] Priya, Jithin Mathews, et al. "A graph theoretical approach for identifying fraudulent transactions in circular trading." DATA ANALYTICS 2017 (2017): 36.
>
> [4] https://www.nebula-graph.io/
>
> [5] https://www.nebula-graph.io/posts/fraud-detection-using-knowledge-and-graph-database

---

> > ### Author Response · Authors · 2023-08-29
> > **Looking for reply**
> >
> > Dear reviewer:
> >
> > We thank you for your review and suggestions. We have made significant modifications to our paper accordingly, adding a large number of new discussions and analyses into our paper (most of them have to be included in the Appendix because of the space limit, and the new material is highlighted in blue color).
> >
> > We hope that our improvement of the paper can address many of your concerns raised in your review and help our paper be understood better, and we are looking forward to your reply.
> >
> > Best regards,
> >
> > Paper346 Authors

---

### Official Review · Reviewer_YhLr · 2023-07-22
**A dataset to evaluate Complex Query Answering (CQA) models**

**Rating:** 5
**Confidence:** 2
**Correctness:** The data set is constructed in a soun…

**Strengths:**

1- The authors provide a dataset that surpasses current benchmark datasets in complexity, offering a valuable resource for more robust evaluations of CQA models.
2- They introduce a framework for query generation and model evaluation, offering a systematic approach for assessing models on diverse queries.

**Additional Feedback:**

N/A

**Clarity:**

The paper contains numerous definitions, but providing more examples alongside them would have been beneficial for better comprehension and understanding of those concepts.

**Documentation:**

URL to the dataset and other necessary information are provided.

**Opportunities For Improvement:**

1- The paper lacks novelty as many ideas seem to be derived from previous studies, leading to a potential lack of significant contributions to the field.
2- Some sections of the paper, such as the statement that existing datasets lack structural hardness, remain unclear and require further elaboration. Additionally, the mention of tree-from queries without any explanation adds confusion to the reader's understanding.
3-The presentation of results should be made more coherent and consistent. For instance, there is no clear explanation provided for the superior performance of the cyclic graph in comparison to SDGA.. Moreover, the rationale behind the decision to include k=1,2, and the use of MRR in Table 1 and Hits@10 in Table 2 needs clarification.

**Relation To Prior Work:**

The inclusion of additional examples and more detailed explanations would have been advantageous in clarifying the differences between the presented dataset and existing ones.

**Summary And Contributions:**

In this paper, the authors present EFO_k- CQA, an extension of EFO-1-QA [27], which introduces 741 types of queries and aims to evaluate Complex Query Answering (CQA) models. They propose a comprehensive framework encompassing query generation, answer sampling, model training, and model evaluation.

---

> ### Author Response · Authors · 2023-08-21
>
> Thank you for your comment, we have modified our paper accordingly, adding a large number of new discussions and analyses to our paper, we welcome you to check our general feedback. However, there are also some misunderstandings we would like to clarify.
>
> > 1- The paper lacks novelty as many ideas seem to be derived from previous studies, leading to a potential lack of significant contributions to the field.
>
> Surely, there are lots of preliminaries in our paper, the whole section 2 takes up two pages, inheriting many previous practices, for example, the way to represent existing queries, however, we believe this is evidence that we align with standard protocol in the field of CQA and succeed to find its connection to the CSP problem that helps us understand CQA from a more theoretical standpoint. This alignment should not be considered non-novel, but is a proof of our soundness.
>
> Moreover, we have summarized the contributions of our paper in our introduction (Line 48-61) and we would like to clarify that those four contributions are all novel and can contribute to the field of CQA.
>
> 1.**Complete coverage** This is theoretically backed up by the whole Section 3 and we welcome you to check Appendix D.3 for a detailed discussion to differentiate our EFOk-CQA from previous ones.
>
> 2.**Curated dataset** This is surely novel, as this is a dataset track and we mainly focus on the construction of the dataset itself.
>
> 3.**Convenient Implementation**. We explain the implementation of our framework in Section 4, and supplement it with Appendix F, G. We include end-to-end machine learning model implementation as well as the query generation and answer sampling. This part is supported by our submitted code and should also be considered novel.
>
> 4.**Results and Findings** In section 5, we conduct new experiments that reveal new empirical findings and highlight the challenges of our newly proposed EFOk-CQA dataset. Therefore, this part is also novel.
>
> In general, we are focusing on a dataset and benchmark track and we believe our contribution to the whole process concerning the dataset construction(1) and corresponding benchmark experiment(2) is novel, as already explained and summarize in our introduction.
>
> > 2- Some sections of the paper, such as the statement that existing datasets lack structural hardness, remain unclear and require further elaboration. Additionally, the mention of tree-from queries without any explanation adds confusion to the reader's understanding.”
>
> The fact that existing tree-form queries are subject to structure-based tractability is not the finding of our paper thus we have not included the explanation in our paper, we have included it as well as the whole introduction of tree form query as additional preliminaries in the appendix now according to your advice, please check the appendix C.
>
>
> > 3-The presentation of results should be made more coherent and consistent. For instance, there is no clear explanation provided for the superior performance of the cyclic graph in comparison to SDGA.. Moreover, the rationale behind the decision to include k=1,2, and the use of MRR in Table 1 and Hits@10 in Table 2 needs clarification.”
>
> We have provided our conjecture of the phenomenon of superior performance of the cyclic graph in Section 5.2, structural analysis paragraph (Line 266-268).  Additionally, we have replaced MRR with HIT@10 in Table 1 now, please check that.
>
> > The inclusion of additional examples and more detailed explanations would have been advantageous in clarifying the differences between the presented dataset and existing ones.
>
>  Thank you for your comment. We have already included more detailed explanations about the difference between our EFOk-CQA dataset with the previous dataset in Appendix D.3 now (mentioned in the last line of Section 3.2). Please check that.

---

> > ### Author Response · Authors · 2023-08-29
> > **Looking forward to reply**
> >
> > Dear reviewer:
> >
> > We thank you for your review and suggestions. We have made significant modifications to our paper accordingly, adding a large number of new discussions and analyses into our paper (most of them have to be included in the Appendix because of the space limit, and the new material is highlighted in blue color).
> >
> > We hope that our improvement of the paper can address many of your concerns raised in your review and help our paper be understood better, and we are looking forward to your reply.
> >
> > Best regards,
> >
> > Paper346 Authors

---

### Official Review · Reviewer_cx54 · 2023-07-23
**Benchmark for query answering on knowledge graphs**

**Rating:** 4
**Confidence:** 3
**Correctness:** Yes. It is based on a formal framewor…

**Strengths:**

- The framework is formally defined and grounded in first order logic. Connections to CSP problems are well articulated.

- The proposed class of queries is more expressive than those proposed in previous work (e.g., can have more than one free variable)

- The dataset of queries released by the authors is useful for researchers in this area.

**Additional Feedback:**

-

**Clarity:**

The paper is well written but would benefit from a better explanation of the intuition behind the problem and its real-world applications

**Documentation:**

Yes

**Ethics:**

No. Only synthetic data is used.

**Limitations:**

As mentioned above, a limitation of this work is that the dataset lacks queries inspired by real-world use cases. I do understand the need for synthetic queries but I would encourage the authors to have a mix of queries, some of them synthetically created but others capturing the expected use cases for this class of queries.

**Opportunities For Improvement:**

- There is no discussion in the paper on real-world use cases or practical motivation for this class of queries. The entire dataset is constructed synthetically and therefore does not help understand if methods help in practical use cases. The authors mention TPC-H in the paper, which is a major benchmark used for relational databases. However, the value of TPC-H is that it provides queries that are representative of real-world use cases (only the database instances are created synthetically). In contrast, this paper creates all queries synthetically.

- The dataset is used to evaluate existing methods under a number of metrics proposed by the authors. However, no clear trends are discovered for queries with more than one free variable (Table 2).

- The paper would be more suitable for the audience of a database or database theory conference than for NeurIPS.



**Relation To Prior Work:**

Yes

**Summary And Contributions:**

The paper introduces a class of queries for query answering on knowledge graphs and a method for creating synthetic queries. A dataset with synthetic queries is created and used to evaluate existing methods.

---

> ### Author Response · Authors · 2023-08-21
>
> Thank you for your review, we have modified our paper accordingly, adding a large number of new discussions and analyses to our paper, we welcome you to check our general feedback. However, there are also some misunderstandings we would like to clarify.
>
>
> > There is no discussion in the paper on real-world use cases or practical motivation for this class of queries .... In contrast, this paper creates all queries synthetically.
>
> The task of complex query answering has real-world applications, like fact ranking[1], and explainable recommendation[2]. However, some important practical applications can not be covered by existing dataset in complex query answering, because their construction is biased and have not discussed some key features of the query.
>
> We would like to introduce **one example in fraud detection** where we need to detect a group of people with cyclic money flow for anti-money laundering applications[3], we also note that this finding is also shared by open-source graph database[4,5]. Therefore, our investigation on cyclic queries and queries with more than one free variable can be justified to help develop more versatile CQA models that are suitable for more real-world applications.
>
> In comparison with the TPC-H which only has dozens of query types, we also believe that our thorough exploration of the whole combinatorial space (in Section 4.1 and detailedly explained in Appendix D.4) is important because our setting is fundamentally different than the TPC-H. The empirical result of a more complex query is hard to predict from other queries because of the open world assumption, as well as the fact that error can accumulate in the machine learning models, different from traditional database settings. Moreover, our thorough exploration indeed may help to detect untrustworthy machine learning models. We have included more discussions regarding this issue, we hope you may check the 5th part of our general rebuttal.
>
>
> > The dataset is used to evaluate existing methods under a number of metrics proposed by the authors. However, no clear trends are discovered for queries with more than one free variable (Table 2).
>
> We have discussed the result of queries with more than two free variables in Section 5.3, where we have three findings that are further corroborated in Appendix F as mentioned in Line 254.
> 1. We test our newly proposed metrics and find that the three kinds of metrics differ vastly on the scale of the final score, specifically, the ultimate goal to answer EFOk queries (joint HIT@10) is extremely low, indicating the great challenge to finally resolve the EFOk queries. (Line 289-292)
> 2. Most of the analysis is similar to our analysis in Section 5.2, where we discuss the experiment result for queries with one free variable, the reason is that all the existing methods are designed solely for inferring EFO1 queries and none of them have considered our EFOk queries with combinatorial answers.(Line 287-289)
> 3. Moreover, we still get some fresh results, the model FIT falls behind other models in the cyclic graph when the queries have two free variables, (Line 292-293) this may be explained by the fact that it degenerates to enumeration in dealing with cyclic graph and does not scale up for more complex queries, exposing its inherent weakness even though it outperforms every other model in queries with one free variable as shown in Table 1.(Line 279-281) We also offered additional discussions in Appendix F as promised in Line 281, and Line 687-695 is the detail.
>
>
> > The paper would be more suitable for the audience of a database or database theory conference than for NeurIPS.
>
> Our work is an interdisciplinary work, that aims to represent database-level queries to serve the complex query answering task which is in the interest of the machine learning community. Most importantly, our work is based on the open world assumption, where the models are required to infer those answers that can not be derived by database algorithms and the neural methods have more scalability in large knowledge graphs. This difference is already in our **introduction**(Line 21-28), and we hope you may check that. Therefore, we not only systematically provide a new way to investigate the EFOk queries (Section 3),  but also provide the whole framework that is able to support end-to-end machine learning model (Section 4).
>
> [1] Ren, Hongyu, et al. "Fact Ranking over Large-Scale Knowledge Graphs with Reasoning Embedding Models." Data Engineering: 124.
>
> [2] Syed, Muzamil Hussain, Tran Quoc Bao Huy, and Sun-Tae Chung. "Context-aware explainable recommendation based on domain knowledge graph." Big Data and Cognitive Computing 6.1 (2022): 11.
>
> [3] Priya, Jithin Mathews, et al. "A graph theoretical approach for identifying fraudulent transactions in circular trading." DATA ANALYTICS 2017 (2017): 36.
>
> [4] https://www.nebula-graph.io/
>
> [5] https://www.nebula-graph.io/posts/fraud-detection-using-knowledge-and-graph-database

---

> > ### Author Response · Authors · 2023-08-29
> > **Looking for reply**
> >
> > Dear reviewer:
> >
> > We thank you for your review and suggestions. We have made significant modifications to our paper accordingly, adding a large number of new discussions and analyses into our paper (most of them have to be included in the Appendix because of the space limit, and the new material is highlighted in blue color).
> >
> > We hope that our improvement of the paper can address many of your concerns raised in your review and help our paper be understood better, and we are looking forward to your reply.
> >
> > Best regards,
> >
> > Paper346 Authors

---

### Author Response · Authors · 2023-08-21
**General feedback about new modification of the paper**

Dear reviewers:

We thank all of you for your review and suggestions. We have modified our paper accordingly, adding a large number of new discussions and analyses into our paper (most of them have to be included in the Appendix because of the space limit), the new material is highlighted in blue color in the new pdf and we want to summarize those modifications to help you better get hold of our improvement.

1. Some typos and unclear expression is addressed in the main paper.

2. We include a new section in Appendix, **Appendix C**, where we formally introduce the definition of tree form query and its origin in previous research as preliminary for someone who is interested and also makes our paper more self-contained.

3. The material of **Appendix D.3** is largely increased, including a newly added Figure 5,  where we discuss the difference between our EFOk-CQA dataset and the previous dataset and benchmark in detail and offers the intuitive representation in Figure 5.

4. We use a new section, **Appendix G**, to discuss whether it is possible to utilize our framework in the more avant-garde complex query answering settings, and we show that some scenarios, such as sampling inductive EFOk query, are within the scope of our framework’s capabilities. Along with that, we also discuss the limitation of our work clearly and present some features that are not contained in our framework, such as n-ary relation.
5. Most importantly, we add more detailed experiment results and analysis in **Appendix H.1**, where we discuss two issues that further help with the development of CQA models: how our EFOk-CQA dataset can help to detect untrustworthy machine learning models, and whether the current dataset is enough to benchmark the performance of current CQA models.

Generally, we have made much effort in improving the soundness, completeness, and clarity. We also provide new discussions about our new empirical findings as well as the future outlook of CQA development. We hope that our improvement of the paper can address many of your concerns raised in your review and help our paper be understood better.

Best regards,

Paper346 Authors

---

### Decision · Program_Chairs · 2023-09-22

**Decision:**

Reject

**Comment:**

In this paper, the authors have proposed   EFO_k- CQA, which introduces 741 types of queries and aims to evaluate Complex Query Answering (CQA) models.  The proposed data set is comprehensive and the developed query framework is complete. However, as stated in the current reviews and discussion,  the  main drawback of the work is that he/ focuses on transductive, conjunctive, entity-centric queries on triple-based graphs,  which is too narrow.